# Genetic insights into resting heart rate and its role in cardiovascular disease

Resting heart rate is associated with cardiovascular diseases and mortality in observational and Mendelian randomization studies. The aims of this study are to extend the number of resting heart rate associated genetic variants and to obtain further insights in resting heart rate biology and its clinical consequences. A genome-wide meta-analysis of 100 studies in up to 835,465 individuals reveals 493 independent genetic variants in 352 loci, including 68 genetic variants outside previously identified resting heart rate associated loci. We prioritize 670 genes and in silico annotations point to their enrichment in cardiomyocytes and provide insights in their ECG signature. Two-sample Mendelian randomization analyses indicate that higher genetically predicted resting heart rate increases risk of dilated cardiomyopathy, but decreases risk of developing atrial fibrillation, ischemic stroke, and cardio-embolic stroke. We do not find evidence for a linear or non-linear genetic association between resting heart rate and all-cause mortality in contrast to our previous Mendelian randomization study. Systematic alteration of key differences between the current and previous Mendelian randomization study indicates that the most likely cause of the discrepancy between these studies arises from false positive findings in previous one-sample MR analyses caused by weak-instrument bias at lower *P*-value thresholds. The results extend our understanding of resting heart rate biology and give additional insights in its role in cardiovascular disease development.

Higher resting heart rate (RHR) is associated with cardiovascular diseases and all-cause mortality in traditional epidemiological studies[1–5]. However, RHR is influenced by disease status and a plethora of potential confounders, which could affect these associations.

A Mendelian randomization (MR) approach, in which genetic variants associated with RHR are used as a proxy for RHR, can also be used to study the association of RHR with cardiovascular diseases and all-cause mortality. Since genetic variants are fixed from conception and then randomly assigned from parents to offspring, they are more immune to reverse causation and confounders[6]. In our previous study, we found evidence for a positive association between genetically predicted RHR and all-cause mortaliy[7]. However, a higher genetically predicted RHR was not found to increase the risk of cardiovascular diseases[7–9] and appeared to decrease the risks of atrial fibrillation and cardio-embolic stroke[9]. Multiple genome-wide association studies (GWAS) have identified RHR-associated genetic variants which could be used as genetic instruments in MR studies to entangle the relationship between RHR and cardiovascular disease[7,8,10]. The two largest GWAS to date have been performed in the UK Biobank[7,10] but were either performed in a subcohort of individuals with available genetic data during its time of publication[7] or lacked a replication cohort[10]. We set out to perform a genome-wide meta-analysis of RHR in the largest sample size to date with internal replication of the associated genetic variants, in order to broaden our knowledge of the biological mechanisms underlying interindividual differences in RHR and identify robustly

e-mail: P.vanderHarst@umcutrecht.nl

associated RHR-associated genetic variants to assess the genetic association of RHR with cardiovascular disease and all-cause mortality.

In this work, a genome-wide meta-analysis of 100 studies in up to 835,465 individuals reveals 493 independent RHR-associated genetic variants in 352 loci, including 68 genetic variants outside previously identified RHR-associated loci (Fig. 1a).

In addition, in silico analysis pinpoint that identified candidate genes are mainly enriched in cardiomyocyte tissue (Fig. 1b), and Mendelian randomization analyses indicate that higher genetically predicted resting heart rate is not associated with all-cause mortality, increases the risk of dilated cardiomyopathy, but decreases the risk of developing atrial fibrillation, ischemic stroke, and cardio-embolic stroke (Fig. 1c).

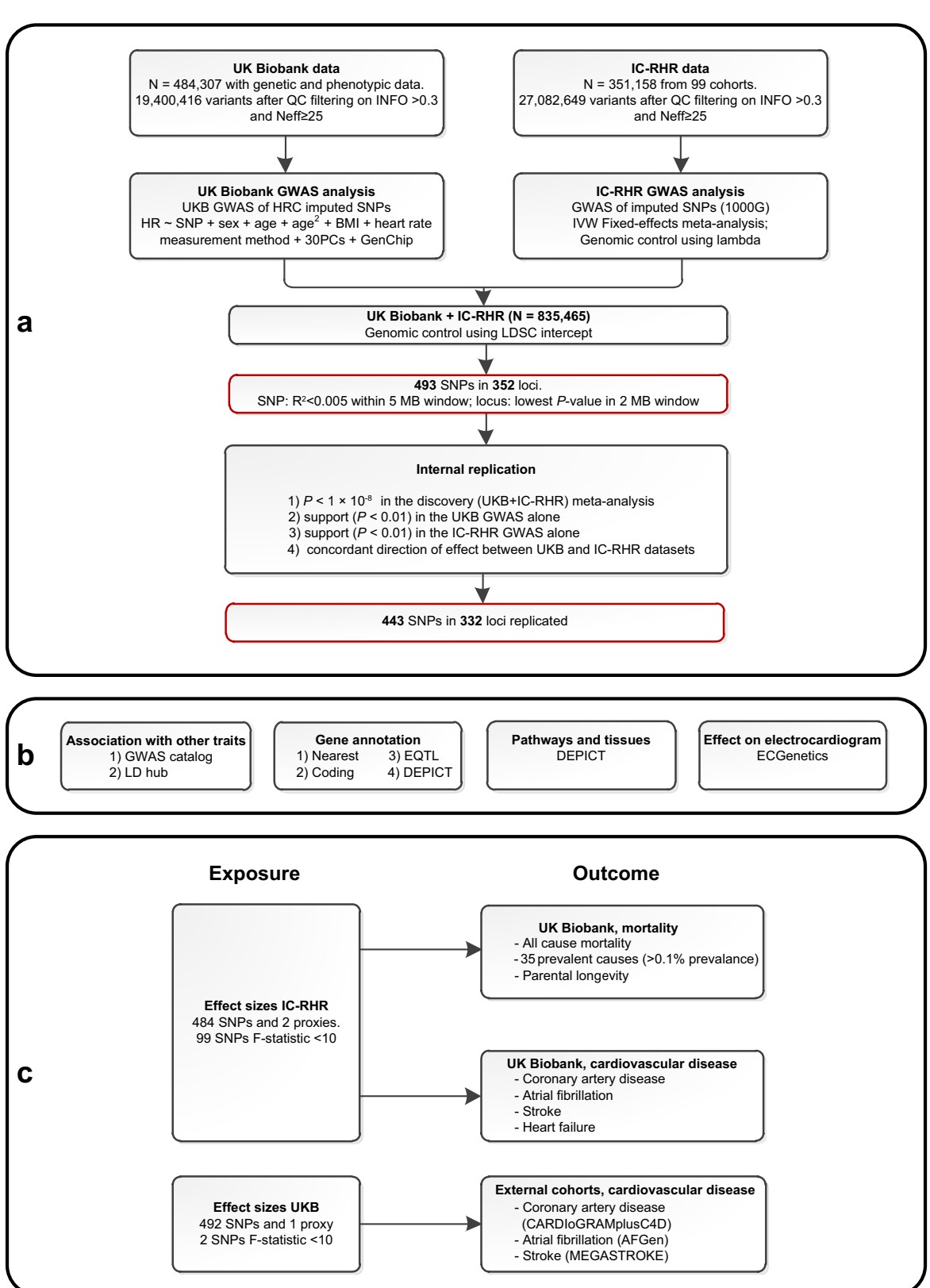

**Fig. 1 | Study flowchart showing the study design, in silico annotations, and functional analyses. a** Schematic overview of the study design for the discovery and replication of genetic loci associated with resting heart rate (RHR) using mixed linear models with a two-sided *P*-value of $P < \times 10^{-8}$ to define genome-wide significance. A genome-wide significant genetic variant was considered replicated if $P < 0.01$ in the UK Biobank and IC-RHR cohort with concordant effect sizes. The black-bordered boxes show the methodology, the red-bordered boxes show the most important results. **b** Analyses performed to evaluate RHR-associated genetic variants and to gain further insights into the underlying biology. **c** Schematic presentation of the two-sample Mendelian randomization analyses of genetically predicted RHR on mortality and cardiovascular diseases. Effect sizes were taken

from the IC-RHR data to test the associations with mortality and cardiovascular diseases in the UK Biobank. Effect sizes were taken from the UK Biobank to test the association with coronary artery disease and myocardial infarction in the CARDIoGRAMplusC4D cohort, atrial fibrillation in the AFGen cohort, and any, ischemic, cardio-embolic, large artery and small vessel stroke within the MEGASTROKE consortium. BMI body mass index, GWAS genome-wide association study, HRC Haplotype Reference Panel, IC-RHR International Consortium for Resting Heart Rate, MB megabase, *N* sample size, Neff effective sample size, PC principal components, RHR resting heart rate, SNPs single nucleotide polymorphisms, QC quality control, 1000G = 1000 Genomes.

## Results

### Genome-wide meta-analysis of resting heart rate

We performed a meta-analysis of RHR GWAS using 99 cohorts consisting of up to 351,158 individuals, which from here on will be referred to as the International Consortium of Resting Heart Rate (IC-RHR). Second, we performed a GWAS on 484,307 individuals from the UK Biobank. These large cohorts were meta-analyzed to include up to 835,464 individuals, in whom 30,458,884 directly genotyped and imputed autosomal variants were analyzed (Supplementary Data 1, Fig. 1a). The meta-analysis revealed 493 independent genetic variants in 352 loci. Out of these 493 independent genetic variants, 68 were outside previously identified RHR-associated loci and 67 of those were internally replicated (Fig. 2a, Supplementary Data 2)[7,8,10]. Out of the 425 genetic variants inside previously identified RHR-associated loci, a total of 376 were internally replicated. In addition, 332 out of the 352 loci were considered internally replicated as they showed the concordant direction of effects and nominal associations ($P < 0.01$) in the UK Biobank GWAS and IC-RHR meta-analysis (Fig. 2b). The RHR-associated genetic variants identified previous studies from Eppinga et al. and Den hoed et al. were all replicated in the current study (Supplementary Data 3). A total of 74 loci identified in the study by Guo et al. were not replicated in the current study, of which 40 would not have been identified as loci using the current GWAS clumping criteria. The remaining 34 loci did not reach genome-wide significance in the current meta-analysis with generally high *P*-values in the IC-RHR consortium, probably therefore failing replication (Supplementary Data 3). The linkage disequilibrium (LD) score regression intercept of the meta-analysis was $1.051 \pm 0.002$, suggesting little evidence of genomic inflation (Fig. 2c). The QQ plots for the UK Biobank GWAS and IC-RHR meta-analysis are shown in Supplementary Fig. 1. The genomic control lambda's, LD-Score intercepts and the attenuation ratio statistics suggested no inflation due to non-polygenic signals[11,12]. Single nucleotide polymorphism (SNP) heritability of RHR, as calculated by LD score regression, was estimated to be 10%. A polygenic score weighted by the effect sizes of the IC-RHR explained 5.33% of the variation in RHR in the UK Biobank. A Chow test indicated the absence of strong differences between participants with a history of any cardiovascular disease or use of RHR-altering medication versus participants without such a medical history (Supplementary Data 4). Genetic correlation analyses were performed and we observed significant correlations with anthropometric measurements, pulse wave reflection index, and physical activity measurements (Supplementary Data 5). A query of the GWAS Catalog showed that the 493 genetic variants associated with RHR were most commonly in high LD (LD > 0.8) with anthropometric measurements and blood pressure traits (Supplementary Data 6).

### Candidate genes and insights into biology

We explored the potential biology of the 352 RHR-associated loci by prioritizing candidate genes in these loci (Supplementary Data 2). A total of 407 unique genes were in close proximity to the lead variant, defined as the nearest gene and any additional gene within 10 kb (Supplementary Data 2). There were 52 genes that contained

coding genetic variants in LD ($R^2 > 0.80$) with RHR lead variants. Functional annotation of these coding variants is provided in Supplementary Data 7. Using summary data-based Mendelian randomization analysis (SMR) and heterogeneity in dependent instruments (HEIDI) tests, we found that the RHR-associated loci and eQTLs colocalized at 88 genes (Supplementary Data 8)[13]. Lastly, 381 unique genes were taken forward by data-driven expression-prioritized integration for complex traits (DEPICT) analyses (Supplementary Data 9). Of the 670 unique candidate causal genes identified, 33 genes were prioritized by at least three out of four established methods, which may be used to prioritize candidate genes (Fig. 2d). Of these genes, *PHACTR4*, *ENO3*, and *SENP2* were prioritized by all four methods. Full annotations of all identified genes are in Supplementary Data 10.

### Pathway analyses and tissue enrichment

Pathway analysis performed by DEPICT showed that RHR revolved around mainly cardiac biology, including cardiac tissue development, muscle cell differentiation, and pro-arrhythmogenic pathways. A total of 1471 reconstituted gene sets within 155 gene clusters were significantly associated with RHR (FDR < 0.05). The newly discovered gene clusters consisted of mostly protein–protein interaction pathways and were commonly located in the periphery of the network (Supplementary Data 11, Supplementary Fig. 2). The tissue enrichment analysis by DEPICT showed 28 tissues at FDR < 0.05 and implicated the cardiovascular system as the most important tissue type, with 8 of the 10 most significantly enriched tissues located within the cardiovascular system (Fig. 3, Supplementary Data 12). Non-cardiovascular tissues with enrichment included muscle and fat tissues, the adrenal glands, the esophagus, and urogenital structures. Conditional analyses showed that associations with non-cardiovascular tissues were rather due to co-expression of RHR genes in cardiovascular tissue than independent enrichment of RHR-associated genes in non-cardiovascular tissues (Fig. 3).

### ECG morphology

The ECGenetics browser, which contains genome-wide summary statistics of every time-point of the complete cardiac cycle at a resolution of 500 Hz, was used to gain insights into the electrophysiological effect of the RHR genetic variants[14]. A total of 86 genetic variants were strongly associated with at least one ECG time point on the non-normalized and normalized association patterns across the full RR interval at a stringent Bonferroni-corrected *P*-value $< 1 \times 10^{-7}$. The associations represented a plethora of ECG morphologies (Supplementary Data 13, Supplementary Fig. 3). The *ACHE*, *ANKRD1*, and *SCN5A* genes exhibited their largest electrical effects on atrial depolarization, *BAG3* and *TTN* on ventricular depolarization, and *RGS6* and *SYT10* on ventricular repolarization. The ECG-wide MR highlighted several loci that had not been associated with resting heart rate or cardiac rhythm and structure previously. The *CCLN1* gene exhibited strong effects on atrial depolarization, *RAP1A* and *ZBTB38* exerted strong effects on early and late ventricular repolarization, respectively. The ECG-wide MR showed that RHR variants exert the largest effect on ventricular

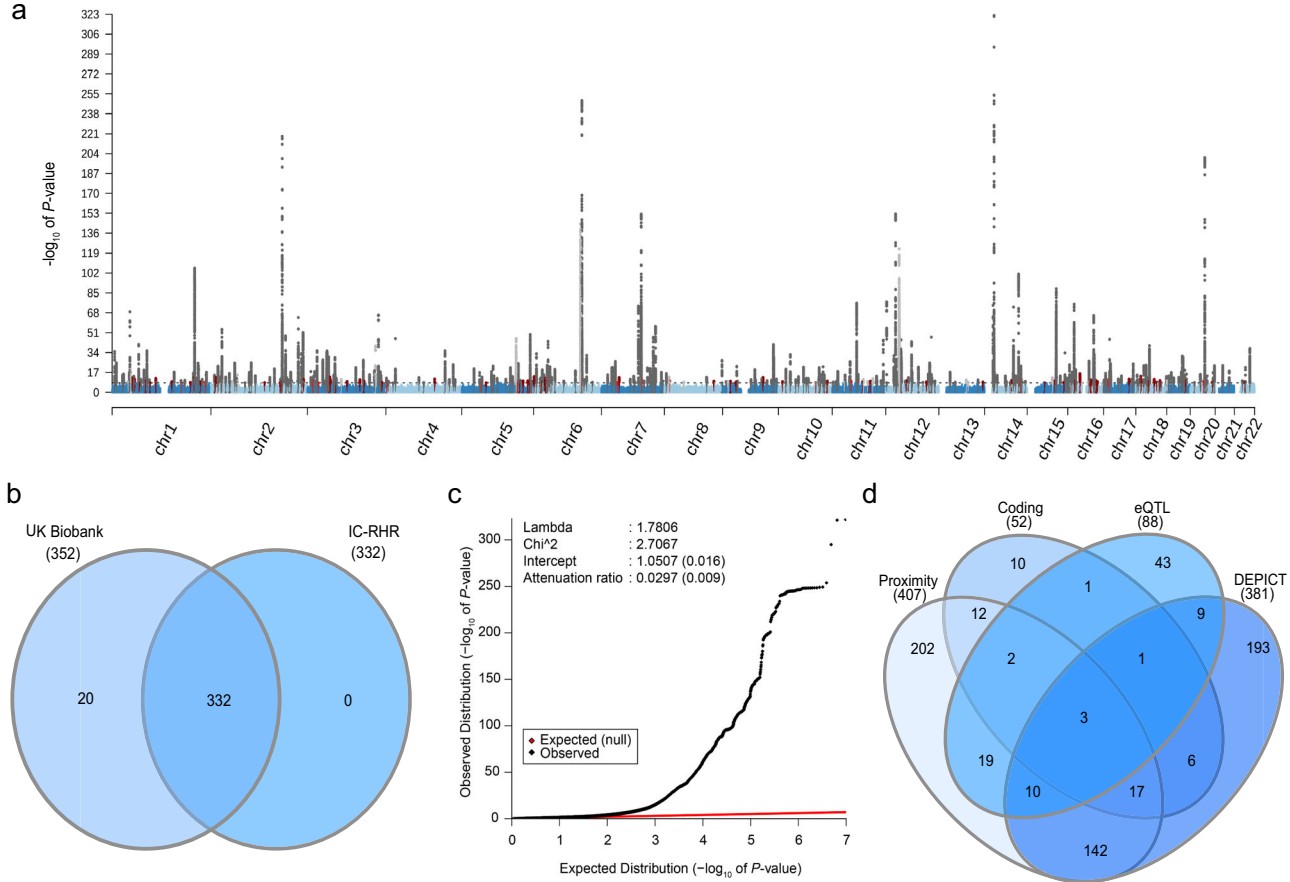

**Fig. 2 | Overview of the findings in the genome-wide association study and in silico search of candidate causal genes. a** Manhattan plot showing the $-\log_{10}(P$-value) for the association of all genotyped or imputed genetics variants with resting heart rate (RHR) assessed using mixed linear models. Red indicates novel and internally replicated RHR-associated loci and black indicates novel but unreplicated RHR-associated loci. Dark gray indicates RHR-associated genetic variants within 1 MB of previously identified RHR-associated loci, which were internally replicated in the current study. Light gray indicates RHR-associated genetic variants within 1 MB of previously identified RHR-associated loci, which were not internally replicated in the current study. A two-sided $P$-value of $P < 1 \times 10^{-8}$ was used to define genome-wide significance. A genome-wide significant genetic variant was considered replicated if $P < 0.01$ in the UK Biobank and IC-RHR cohort with concordant effect sizes. **b** Venn diagram of the 352 identified loci. Of the 352 loci, 332 were internally replicated. **c** Quantile–quantile (QQ) plot of the final meta-analysis. The

black dots represent the observed statistic for the genotyped genetic variants against the corresponding expected statistic. The linkage disequilibrium score regression intercept after the final meta-analysis was 1.051, suggesting little evidence of genomic inflation due to non-polygenic signal. **d** Venn diagram of the prioritization of the 670 unique candidate causal genes as identified by one or multiple strategies. Venn plot shows the overlap of genes tagged by one or multiple strategies, including (1) by proximity, the nearest gene or any gene within 10 kb; (2) genes containing coding variants in LD with RHR-associated variants at $R^2 > 0.8$; (3) eQTL genes in LD ($R^2 > 0.8$) with RHR-associated variants which achieved a Bonferroni corrected two-sided $P = 2.65 \times 10^{-7}$ and passed the HEIDI test at a $P > 0.05$; and (4) DEPICT genes which achieved multiple hypotheses corrected FDR < 0.05. DEPICT data-driven expression prioritized integration for complex traits, eQTL expression quantitative trait loci.

repolarization on both the non-normalized and normalized association patterns (Supplementary Fig. 4).

**Single-nucleus RNA expression**

Single-nucleus RNA sequencing data obtained from a healthy human heart revealed that RHR gene expression is highest in ventricular cardiomyocytes, followed by atrial cardiomyocytes (Supplementary Data 14)[15]. The candidate genes of genetic variants involved in non-isoelectric parts of the ECG showed stronger expression patterns than the isoelectric parts, for example, those involved in left atrial depolarization (*ANKRD1*), ventricular depolarization (*FOHD3, RBM20, MYO18B, TTN*) and ventricular repolarization (*CACNA1C*) (Supplementary Fig. 3).

**Mendelian randomization analyses**

Series of two-sample MR analyses were performed to test whether genetically predicted RHR is associated with all-cause mortality and cardiovascular diseases (definitions provided in Supplementary

Data 15, 16). We initially used the inverse variance weighted multiplicative random-effects (IVW-MRE) model, which provides a consistent estimate under the assumption of balanced pleiotropy. If we found evidence for a genetic association using the IVW-MRE model, we further interrogated these findings using several sensitivity analyses that are more robust to different sources of bias in MR analyses.

First, we assessed the association between genetically predicted RHR and all cause-mortality in the UK Biobank participants over a median follow-up of 8.9 years (interquartile range 8.2–9.5) (Supplementary Data 17–19). Genetically predicted RHR was not associated with the risk of all-cause mortality (HR 1.024, 95% CI 0.993–1.057, $P = 0.13$), as shown in Fig. 4a. We did not find evidence for an association between genetically predicted RHR and parental longevity. Neither did we find evidence for an association between RHR and the 35 leading causes of mortality in the UK Biobank. Systematic alteration of key differences between the current and previous Mendelian randomization study indicated that the most likely cause of the discrepancy between these studies arises from false positive findings in

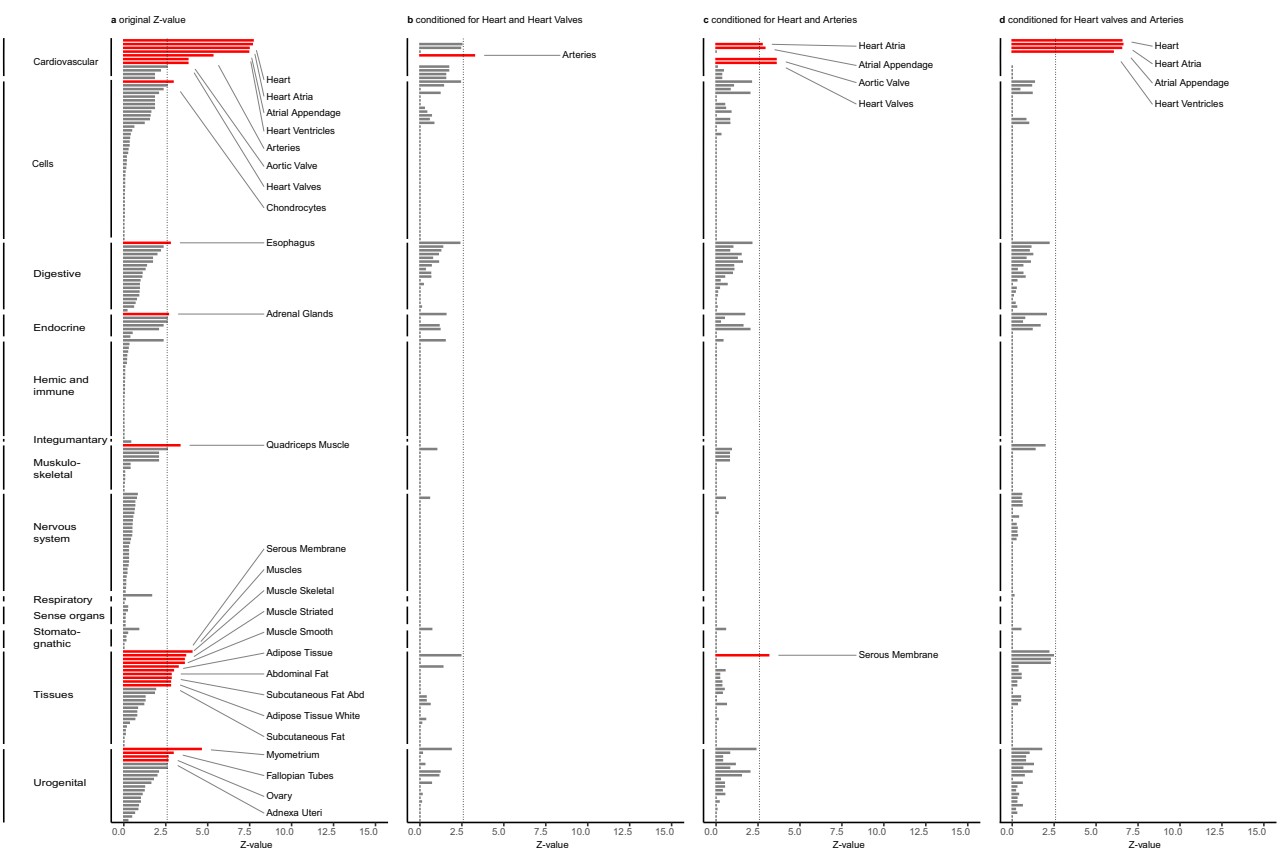

**Fig. 3 | Conditional analyses of tissue enrichment by DEPICT emphasizes cardiac tissue for RHR biology. a** Results of the DEPICT tissue enrichment analysis. The *Y*-axis shows the tissues clustered by the first MeSH term, ordered on *Z*-value per cluster. The *X*-axis shows the *Z*-value. A multiple comparisons corrected two-sided FDR < 0.05, corresponding to a *P*-value < 9.75 × 10⁻³ and *Z*-value of 2.585, was considered to be statistically significant. Significant tissues are plotted in red and annotated, other tissues are plotted in gray. Conditional analyses were performed by correcting for the tissue with the highest *Z*-value to investigate whether significant tissues were independently associated with RHR. Not a single tissue remained significant at an FDR < 0.05 after three consecutive corrections (for the heart, heart valve, and arteries). Panel **b**–**d** shows *Z*-values of all tissues after consecutive correction for respectively heart and heart valves, heart and arteries, and heart valve and arteries. This jointly provides information on which the other tissues are co-dependent. FDR false discovery rate.

previous one-sample MR analyses caused by weak-instrument bias at lower *P*-value thresholds (Supplementary Fig. 5, Supplementary Data 20)[7]. Non-linear MR analysis showed an always-increasing dose–response relation between genetically predicted RHR and all-cause mortality that was compatible with a null effect (Fig. 4b, Supplementary Data 21, 22), providing no evidence for a U-shaped pattern that has been previously described[3].

We then explored the association between genetically predicted RHR and several prevalent cardiovascular diseases. We did not find evidence for an association between genetically predicted RHR and coronary artery disease in the UK Biobank (OR 0.977, 95% CI 0.946–1.009, *P* = 0.16) or in the CARDIoGRAMplusC4D cohort (OR 0.976, 95% CI 0.944–1.010, *P* = 0.17), in line with previous analyses[8,9]. Similarly, there was no evidence for an association between genetically predicted RHR and myocardial infarction in the UK Biobank or in the CARDIoGRAMplusC4D cohort (Fig. 5, Supplementary Data 23–26). We found no evidence for non-linear dose–response relations of genetically predicted RHR with coronary artery disease or myocardial infarction (Supplementary Data 21, 22).

Higher genetically predicted RHR was suggestively associated with a lower risk of atrial fibrillation development in the UK Biobank (OR 0.946, 95% CI 0.897–0.998, *P* = 0.04) and in the AFgen consortium (OR 0.942, 95% CI 0.897–0.989, *P* = 0.02), but these results were not significant after correction for multiple testing (*P* < 4.17 × 10⁻³). MR-Lasso, which can provide evidence for potential causal associations when there is a small number of genetic variants with heterogeneous

ratio estimates, indicated that genetically predicted RHR was significantly inversely associated with atrial fibrillation (Fig. 5, Supplementary Data 23). The contamination mixture model, which provides evidence for potential causal associations if the plurality of the genetic instruments is valid, provided evidence for a negative association between genetically predicted RHR and atrial fibrillation in the UK Biobank cohort, but this was not replicated in the AFgen consortium (Fig. 5). Non-linear MR analyses showed a significant negative exponential growth pattern in the dose–response relation between genetically predicted RHR and atrial fibrillation (Supplementary Data 21, 22, Supplementary Fig. 8). Specifically, individuals at the extreme right tail of the distribution of genetically predicted RHR had a lower risk of atrial fibrillation. For example, compared with the population mean RHR of approximately 70 bpm, individuals with a genetically predicted RHR of 89 and 98 bpm had a significantly lower risk of atrial fibrillation (OR 0.969, 95% CI 0.941–0.998, *P* = 0.04; OR 0.922, 95% CI 0.897–0.948, *P* = 6.36 × 10⁻⁹), while this was not true for a genetically predicted RHR of 80 bpm (OR 1.000, 95% CI 0.968–1.034, *P* = 0.99).

We found that higher genetically predicted RHR is associated with the risk of any stroke (OR 0.951, 95% CI 0.926–0.976, *P* = 1.59 × 10⁻⁴), ischemic stroke (OR 0.940, 95% CI 0.915–0.967, *P* = 1.08 × 10⁻⁵) and cardio-embolic stroke (OR 0.875, 95% CI 0.828–0.925, *P* = 2.11 × 10⁻⁶), suggestively associated with large artery stroke (OR 0.939, 95% CI 0.884–0.998, *P* = 0.04) and not with small vessel stroke (OR 1.001, 95% CI 0.950–1.055, *P* = 0.97) in the MEGASTROKE consortium. The results were consistent across MR methods for any, ischemic and

**a** Forestplot of the linear MR between RHR and all−cause mortality

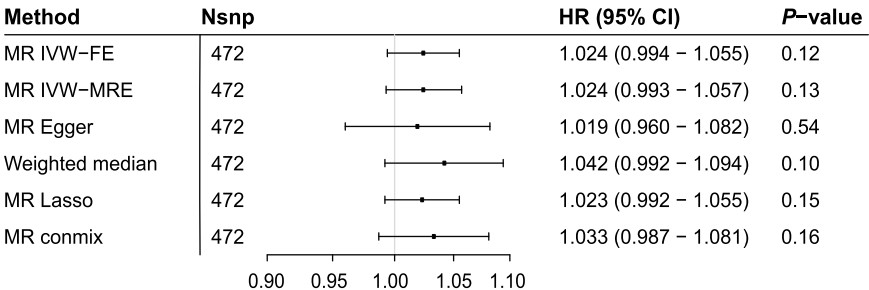

| Method | Nsnp | | HR (95% CI) | *P*−value |
|---|---|---|---|---|
| MR IVW−FE | 472 | | 1.024 (0.994 − 1.055) | 0.12 |
| MR IVW−MRE | 472 | | 1.024 (0.993 − 1.057) | 0.13 |
| MR Egger | 472 | | 1.019 (0.960 − 1.082) | 0.54 |
| Weighted median | 472 | | 1.042 (0.992 − 1.094) | 0.10 |
| MR Lasso | 472 | | 1.023 (0.992 − 1.055) | 0.15 |
| MR conmix | 472 | | 1.033 (0.987 − 1.081) | 0.16 |

**b** Dose response curve of the non−linear MR between RHR and all−cause mortality

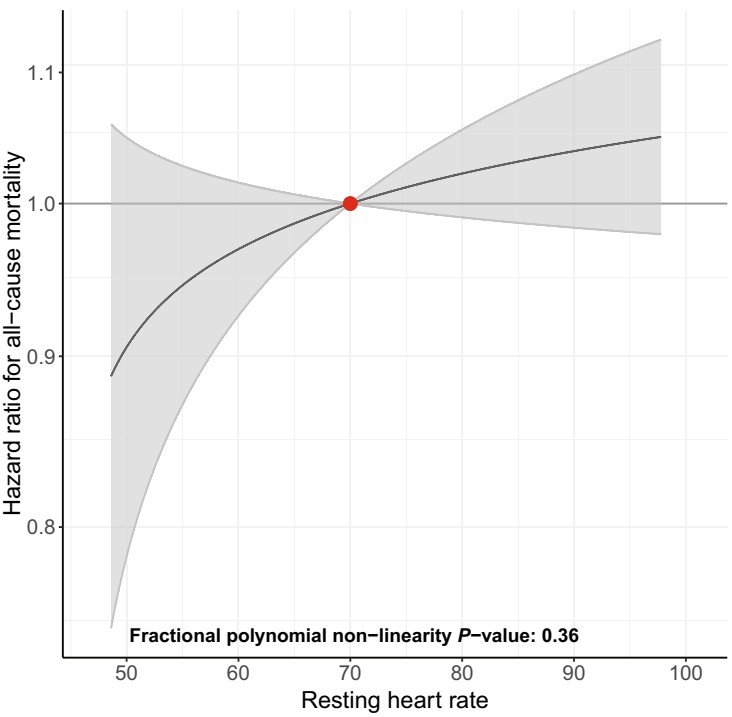

**Fig. 4 | Mendelian randomization shows absence of linear and non-linear associations between genetically predicted RHR and all-cause mortality.** Linear and non-linear Mendelian randomization analyses were performed to test the association between genetically predicted RHR and all-cause mortality. **a** Forest plot of the linear MR analyses between genetically predicted RHR and all-cause mortality (Ncases = 16,289, Ncontrol = 396,183). With a single outcome, a two-sided *P*-value of *P* < 0.05 was considered significant. Hazard ratios and 95% confidence intervals are shown. The *X*-axis shows hazard ratio's on a log$_{10}$ scale, the center as indicated by a gray line depicts a hazard ratio of 1. **b** Dose−response curve of the non-linear MR analyses between genetically predicted RHR and all-cause mortality (Ncases = 16,039, Ncontroles = 394,144). The comparisons are conducted within strata and therefore the graph provides information on the expected average change in the outcome if a person with an RHR of (say) 70 bpm instead had an RHR value of 90 bpm. The gradient at each point of the curve is the localized average causal effect. Shaded areas represent 95% confidence intervals. With a single outcome, a two-sided fractional polynomial non-linearity *P*-value of *P* < 0.05 was considered significant. The *X*-axis shows RHR, the *Y*-axis shows hazard ratios. The center as indicated by a dark gray line depicts a hazard ratio of 1. RHR resting heart rate, HR hazard ratio, CI confidence interval, MR Mendelian randomization, IVW inverse variance weighted, FE fixed effects, MRE multiplicative random effects.

cardioembolic stroke (Fig. 5, Supplementary Data 23). The associations between genetically determined RHR and any stroke or ischemic stroke could not be replicated in the UK Biobank using a univariable MR IVW-MRE approach (OR 0.987, 95% CI 0.953–1.023, *P* = 0.49; OR 0.970, 95% CI 0.928–1.015, *P* = 0.19). We found no evidence for a non-linear association between genetically determined RHR and any ischemic stroke (Supplementary Data 21, 22). We used a multivariable MR approach to gain insights into potential mediating factors or pleiotropic pathways in the association between genetically predicted RHR and stroke. First, we found that the direct effects of RHR on cardio-embolic stroke to be attenuated by the effects of atrial

fibrillation and estimated the attenuation through atrial fibrillation to be 18.4% (Fig. 6, Supplementary Data 27, 28). The direct effects of genetically predicted RHR on any stroke and ischemic stroke were most strongly attenuated by pulse pressure, with estimated attenuation of 28.1% and 31.5%, respectively (Fig. 6a, b, Supplementary Data 27, 28). There was a strong association between genetically predicted RHR and pulse pressure (*β* = −0.192, SE = 0.019, *P* = 1.81 × 10⁻²⁴), but MR-Steiger sensitivity analysis filtered a large part of the genetic variants and repeating the MR on the remaining subset did not show a significant association between RHR and pulse pressure (*β* = −0.005, SE = 0.008, *P* = 0.57, Supplementary Data 29, 30).

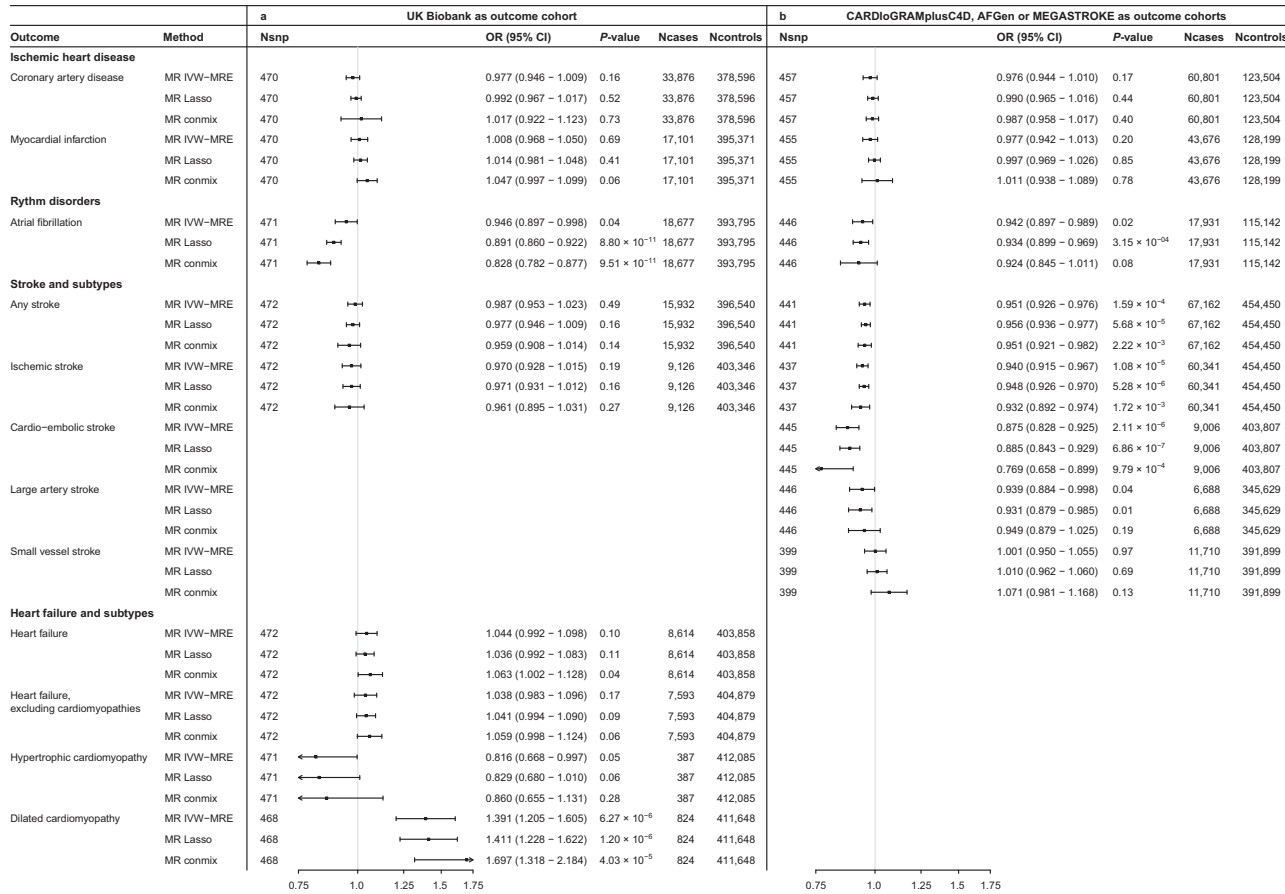

| Outcome | Method | a UK Biobank as outcome cohort |  |  |  |  | b CARDIoGRAMplusC4D, AFGen or MEGASTROKE as outcome cohorts |  |  |  |  |
|---|---|---|---|---|---|---|---|---|---|---|---|
|  |  | Nsnp | OR (95% CI) | P-value | Ncases | Ncontrols | Nsnp | OR (95% CI) | P-value | Ncases | Ncontrols |
| **Ischemic heart disease** | | | | | | | | | | | |
| Coronary artery disease | MR IVW–MRE | 470 | 0.977 (0.946 – 1.009) | 0.16 | 33,876 | 378,596 | 457 | 0.976 (0.944 – 1.010) | 0.17 | 60,801 | 123,504 |
|  | MR Lasso | 470 | 0.992 (0.967 – 1.017) | 0.52 | 33,876 | 378,596 | 457 | 0.990 (0.965 – 1.016) | 0.44 | 60,801 | 123,504 |
|  | MR conmix | 470 | 1.017 (0.922 – 1.123) | 0.73 | 33,876 | 378,596 | 457 | 0.987 (0.958 – 1.017) | 0.40 | 60,801 | 123,504 |
| Myocardial infarction | MR IVW–MRE | 470 | 1.008 (0.968 – 1.050) | 0.69 | 17,101 | 395,371 | 455 | 0.977 (0.942 – 1.013) | 0.20 | 43,676 | 128,199 |
|  | MR Lasso | 470 | 1.014 (0.981 – 1.048) | 0.41 | 17,101 | 395,371 | 455 | 0.997 (0.969 – 1.026) | 0.85 | 43,676 | 128,199 |
|  | MR conmix | 470 | 1.047 (0.997 – 1.099) | 0.06 | 17,101 | 395,371 | 455 | 1.011 (0.938 – 1.089) | 0.78 | 43,676 | 128,199 |
| **Rythm disorders** | | | | | | | | | | | |
| Atrial fibrillation | MR IVW–MRE | 471 | 0.946 (0.897 – 0.998) | 0.04 | 18,677 | 393,795 | 446 | 0.942 (0.897 – 0.989) | 0.02 | 17,931 | 115,142 |
|  | MR Lasso | 471 | 0.891 (0.860 – 0.922) | $8.80 \times 10^{-11}$ | 18,677 | 393,795 | 446 | 0.934 (0.899 – 0.969) | $3.15 \times 10^{-04}$ | 17,931 | 115,142 |
|  | MR conmix | 471 | 0.828 (0.782 – 0.877) | $9.51 \times 10^{-11}$ | 18,677 | 393,795 | 446 | 0.924 (0.845 – 1.011) | 0.08 | 17,931 | 115,142 |
| **Stroke and subtypes** | | | | | | | | | | | |
| Any stroke | MR IVW–MRE | 472 | 0.987 (0.953 – 1.023) | 0.49 | 15,932 | 396,540 | 441 | 0.951 (0.926 – 0.976) | $1.59 \times 10^{-4}$ | 67,162 | 454,450 |
|  | MR Lasso | 472 | 0.977 (0.946 – 1.009) | 0.16 | 15,932 | 396,540 | 441 | 0.956 (0.936 – 0.977) | $5.68 \times 10^{-5}$ | 67,162 | 454,450 |
|  | MR conmix | 472 | 0.959 (0.908 – 1.014) | 0.14 | 15,932 | 396,540 | 441 | 0.951 (0.921 – 0.982) | $2.22 \times 10^{-3}$ | 67,162 | 454,450 |
| Ischemic stroke | MR IVW–MRE | 472 | 0.970 (0.928 – 1.015) | 0.19 | 9,126 | 403,346 | 437 | 0.940 (0.915 – 0.967) | $1.08 \times 10^{-5}$ | 60,341 | 454,450 |
|  | MR Lasso | 472 | 0.971 (0.931 – 1.012) | 0.16 | 9,126 | 403,346 | 437 | 0.948 (0.926 – 0.970) | $5.28 \times 10^{-6}$ | 60,341 | 454,450 |
|  | MR conmix | 472 | 0.961 (0.895 – 1.031) | 0.27 | 9,126 | 403,346 | 437 | 0.932 (0.892 – 0.974) | $1.72 \times 10^{-3}$ | 60,341 | 454,450 |
| Cardio–embolic stroke | MR IVW–MRE |  |  |  |  |  | 445 | 0.875 (0.828 – 0.925) | $2.11 \times 10^{-6}$ | 9,006 | 403,807 |
|  | MR Lasso |  |  |  |  |  | 445 | 0.885 (0.843 – 0.929) | $6.86 \times 10^{-7}$ | 9,006 | 403,807 |
|  | MR conmix |  |  |  |  |  | 445 | 0.769 (0.658 – 0.899) | $9.79 \times 10^{-4}$ | 9,006 | 403,807 |
| Large artery stroke | MR IVW–MRE |  |  |  |  |  | 446 | 0.939 (0.884 – 0.998) | 0.04 | 6,688 | 345,629 |
|  | MR Lasso |  |  |  |  |  | 446 | 0.931 (0.879 – 0.985) | 0.01 | 6,688 | 345,629 |
|  | MR conmix |  |  |  |  |  | 446 | 0.949 (0.879 – 1.025) | 0.19 | 6,688 | 345,629 |
| Small vessel stroke | MR IVW–MRE |  |  |  |  |  | 399 | 1.001 (0.950 – 1.055) | 0.97 | 11,710 | 391,899 |
|  | MR Lasso |  |  |  |  |  | 399 | 1.010 (0.962 – 1.060) | 0.69 | 11,710 | 391,899 |
|  | MR conmix |  |  |  |  |  | 399 | 1.071 (0.981 – 1.168) | 0.13 | 11,710 | 391,899 |
| **Heart failure and subtypes** | | | | | | | | | | | |
| Heart failure | MR IVW–MRE | 472 | 1.044 (0.992 – 1.098) | 0.10 | 8,614 | 403,858 | | | | | |
|  | MR Lasso | 472 | 1.036 (0.992 – 1.083) | 0.11 | 8,614 | 403,858 | | | | | |
|  | MR conmix | 472 | 1.063 (1.002 – 1.128) | 0.04 | 8,614 | 403,858 | | | | | |
| Heart failure, excluding cardiomyopathies | MR IVW–MRE | 472 | 1.038 (0.983 – 1.096) | 0.17 | 7,593 | 404,879 | | | | | |
|  | MR Lasso | 472 | 1.041 (0.994 – 1.090) | 0.09 | 7,593 | 404,879 | | | | | |
|  | MR conmix | 472 | 1.059 (0.998 – 1.124) | 0.06 | 7,593 | 404,879 | | | | | |
| Hypertrophic cardiomyopathy | MR IVW–MRE | 471 | 0.816 (0.668 – 0.997) | 0.05 | 387 | 412,085 | | | | | |
|  | MR Lasso | 471 | 0.829 (0.680 – 1.010) | 0.06 | 387 | 412,085 | | | | | |
|  | MR conmix | 471 | 0.860 (0.655 – 1.131) | 0.28 | 387 | 412,085 | | | | | |
| Dilated cardiomyopathy | MR IVW–MRE | 468 | 1.391 (1.205 – 1.605) | $6.27 \times 10^{-6}$ | 824 | 411,648 | | | | | |
|  | MR Lasso | 468 | 1.411 (1.228 – 1.622) | $1.20 \times 10^{-6}$ | 824 | 411,648 | | | | | |
|  | MR conmix | 468 | 1.697 (1.318 – 2.184) | $4.03 \times 10^{-5}$ | 824 | 411,648 | | | | | |

**Fig. 5 | Mendelian randomization of genetically predicted RHR on cardiovascular diseases.** Forestplots of the linear Mendelian randomization analyses of resting heart rate (RHR) on cardiovascular diseases. Effect sizes were taken from the IC-RHR data to test the associations with mortality and cardiovascular diseases in the UK Biobank (panel **a**). Effect sizes were taken from the UK Biobank to test the association with cardiovascular diseases in the CARDIoGRAMplusC4D, AFGen, and MEGASTROKE consortia (panel **b**). Results of the MR-IVW, outlier-robust MR-Lasso, and plurality valid MR-Mix are provided. Sample sizes vary per outcome and per cohort and are shown in the figure. Odds ratios and 95% confidence intervals are shown. The X-axis shows odds ratio's on a $\log_{10}$ scale, the center as indicated by a gray line depicts an odds ratio of 1. RHR resting heart rate, MR Mendelian randomization, IVW inverse variance weighted multiplicative random effects, OR odds ratio, CI confidence interval.

Lastly, we found that genetically predicted RHR is associated with an increased risk of dilated cardiomyopathy in the UK Biobank (OR 1.391, 95% CI 1.205–1.605, $P = 6.27 \times 10^{-6}$). The results were robust to MR-Lasso (OR 1.411, 95% CI 1.228–1.622, $P = 1.20 \times 10^{-6}$) and MR-contamination mixture models (OR 1.697, 95% CI 1.318–2.184, $P = 4.03 \times 10^{-5}$). We excluded 96 variants associated with the Q-R upslope at −18 ms of the R peak ($P < 0.05$), which has been established as a biomarker for dilated cardiomyopathy[14], to investigate whether reversed causation contributed to the association with dilated cardiomyopathy. The results were similar to the main analyses (Supplementary Table 1). We did not find evidence for an association between genetically predicted RHR and heart failure, heart failure excluding cardiomyopathies, and hypertrophic cardiomyopathy (Fig. 5, Supplementary Data 23). We did not find evidence for a non-linear association between genetically determined RHR and any type of heart failure (Supplementary Data 21, 22). Scatterplots and dose-response curves of the association between RHR and all assessed outcomes can be found in Supplementary Figs. 6–10.

We assessed whether the Wald estimates between the RHR-associated genetic variants and the cardiovascular diseases could identify risk loci not anticipated to be associated with these outcomes in the outcome GWASs. The locus *FOXC1* for coronary artery disease, *USP39* for myocardial infarction, and *SLC35F1* and *SSPN* for atrial fibrillation were significantly ($P < 1.01 \times 10^{-4}$) and concordantly associated with their respective outcomes in both cohorts while not reaching genome-wide significance in either one of the outcome cohorts (Supplementary Data 31).

## Discussion

We report 493 genetic variants in 352 loci associated with RHR, discovered in the largest GWAS meta-analysis of RHR to date in up to 835,465 individuals[7,8,16,17]. This increase in sample size allowed us to report 68 novel RHR-associated genetic variants and, importantly, provide internal replication for 376 genetic variants previously associated with RHR. A total of 670 candidate genes were prioritized, providing a comprehensive data catalog for future studies on RHR and offering potential new insights into its biology. Four strategies were employed to prioritize candidate genes and *PHACTR4*, *ENO3*, and *SENP2* were highlighted by all four strategies. The *PHACTR4* gene regulates protein phosphatase 1 which interacts with actin and is involved in processes ranging from angiogenesis to cell cycle regulation[18]. It has been associated with pulse pressure and systolic blood pressure in a previous GWAS analysis[19]. *ENO3* encodes beta-enolase, which plays an important role in glycolysis and striated muscle development[20]. It has been implicated in cardiac myocyte development through its function in energy metabolism in both humans and rats[21,22]. *SENP2* encodes sentrin-specific protease 2, which deconjugates small ubiquitin-related modifiers 1 and 2 that are

**a** Multivariable MR of RHR on any stroke (Ncases = 67,162; Ncontrols = 454,450)

| Method | Secondary exposure atrial fibrillation Percentage change = 11.7% | | Secondary exposure systolic blood pressure Percentage change = -2.6% | | Secondary exposure diastolic blood pressure Percentage change = -0.4% | | Secondary exposure pulse pressure Percentage change = 28.1% | |
|---|---|---|---|---|---|---|---|---|
| | Nsnp | P-value | Nsnp | P-value | Nsnp | P-value | Nsnp | P-value |
| Univariable IVW | 390 | $2.36 \times 10^{-4}$ | 372 | $7.66 \times 10^{-4}$ | 354 | $3.94 \times 10^{-4}$ | 372 | $7.66 \times 10^{-4}$ |
| MV IVW | 390 | $1.36 \times 10^{-3}$ | 372 | $5.24 \times 10^{-4}$ | 354 | $4.05 \times 10^{-4}$ | 372 | 0.02 |
| MV MR Egger | 390 | $7.53 \times 10^{-3}$ | 372 | $6.28 \times 10^{-3}$ | 354 | 0.02 | 372 | 0.03 |
| MV MR Median | 390 | $3.90 \times 10^{-3}$ | 372 | $4.65 \times 10^{-3}$ | 354 | $1.88 \times 10^{-3}$ | 372 | 0.14 |
| MV MR Lasso | 390 | $6.54 \times 10^{-4}$ | 372 | $2.47 \times 10^{-5}$ | 354 | $3.13 \times 10^{-5}$ | 372 | $1.59 \times 10^{-3}$ |

**b** Multivariable MR of RHR on any ischemic stroke (Ncases = 60,341; Ncontrols = 454,450)

| Method | Secondary exposure atrial fibrillation Percentage change = 10.5% | | Secondary exposure systolic blood pressure Percentage change = -1.6% | | Secondary exposure diastolic blood pressure Percentage change = -0.1% | | Secondary exposure pulse pressure Percentage change = 28.1% | |
|---|---|---|---|---|---|---|---|---|
| | Nsnp | P-value | Nsnp | P-value | Nsnp | P-value | Nsnp | P-value |
| Univariable IVW | 390 | $3.42 \times 10^{-5}$ | 373 | $1.19 \times 10^{-4}$ | 354 | $1.04 \times 10^{-4}$ | 373 | $1.19 \times 10^{-4}$ |
| MV IVW | 390 | $2.38 \times 10^{-5}$ | 373 | $7.99 \times 10^{-5}$ | 354 | $9.68 \times 10^{-5}$ | 373 | $9.14 \times 10^{-3}$ |
| MV MR Egger | 390 | $1.83 \times 10^{-3}$ | 373 | $1.13 \times 10^{-3}$ | 354 | $4.12 \times 10^{-3}$ | 373 | 0.01 |
| MV MR Median | 390 | $3.18 \times 10^{-4}$ | 373 | 0.02 | 354 | $6.94 \times 10^{-3}$ | 373 | 0.06 |
| MV MR Lasso | 390 | $3.38 \times 10^{-4}$ | 373 | $3.67 \times 10^{-6}$ | 354 | $5.50 \times 10^{-6}$ | 373 | $8.77 \times 10^{-4}$ |

**c** Multivariable MR of RHR on cardio-embolic stroke (Ncases = 9,006; Ncontrols = 403,807)

| Method | Secondary exposure atrial fibrillation Percentage change = 18.4% | | Secondary exposure systolic blood pressure Percentage change = 0.3% | | Secondary exposure diastolic blood pressure Percentage change = 2.1% | | Secondary exposure pulse pressure Percentage change = -5.6% | |
|---|---|---|---|---|---|---|---|---|
| | Nsnp | P-value | Nsnp | P-value | Nsnp | P-value | Nsnp | P-value |
| Univariable IVW | 398 | $6.81 \times 10^{-6}$ | 377 | $4.85 \times 10^{-5}$ | 356 | $6.19 \times 10^{-5}$ | 378 | $4.93 \times 10^{-5}$ |
| MV IVW | 398 | $1.78 \times 10^{-4}$ | 377 | $5.01 \times 10^{-5}$ | 356 | $1.09 \times 10^{-5}$ | 378 | $1.64 \times 10^{-5}$ |
| MV MR Egger | 398 | 0.01 | 377 | $2.81 \times 10^{-3}$ | 356 | 0.02 | 378 | $9.25 \times 10^{-4}$ |
| MV MR Median | 398 | $1.95 \times 10^{-3}$ | 377 | 0.01 | 356 | 0.03 | 378 | $2.09 \times 10^{-3}$ |
| MV MR Lasso | 398 | $3.84 \times 10^{-4}$ | 377 | $2.08 \times 10^{-5}$ | 356 | $1.56 \times 10^{-4}$ | 378 | $2.39 \times 10^{-5}$ |

**Fig. 6 | Multivariable Mendelian randomization reveals pulse pressure and atrial fibrillation as potential mediators of the association of genetically predicted RHR with ischemic and cardio-embolic stroke, respectively.** Forestplots of the results of the two-sample multivariable Mendelian randomization analyses of resting heart rate on **a** any stroke (Ncases = 67,162; Ncontrols = 454,450), **b** ischemic stroke (Ncases = 60,341; Ncontrols = 454,450) and **c** cardio-embolic stroke (Ncases = 9006; Ncontrols = 403,807), when using atrial fibrillation, systolic, diastolic and pulse pressure as secondary exposures. Shown in red are the univariable Mendelian randomization estimates which represent the total estimates of resting heart rate on the outcome. In black are the multivariable Mendelian randomization estimates, which show the direct effect of RHR when corrected for the secondary exposure. These results indicate that atrial fibrillation attenuates the beneficial effect of a higher resting heart rate on cardio-embolic stroke, while pulse pressure attenuates the beneficial effect on any ischemic stroke. MR-Steiger sensitivity analysis indicated that the association between the RHR-associated genetic variants and pulse pressure is unlikely mediated through RHR entirely and biological pleiotropic effects are therefore more likely to cause the attenuation of the association between RHR and stroke when correcting for pulse pressure. Odds ratios and 95% confidence intervals are shown. The X-axis shows odds ratio's on a $\log_{10}$ scale, the center as indicated by a gray line depicts an odds ratio of 1. RHR resting heart rate; MV multivariable, Nsnp number of SNPs.

involved in regulating posttranslational modification of a wide variety of proteins that affect a multitude of different cellular processes[23,24]. Several of these affected proteins are critical in cardiac development and mouse models have shown that alterations in *SENP2* activity lead to congenital heart defects[25,26]. Involvement of *SENP2* in a multitude of cellular processes is reflected by its implication in GWAS of various conditions, including systolic and diastolic blood pressure[27], type 2 diabetes[28], the conduction system of the heart[29,30] and estimated glomerular filtration rate[31]. The loci we associated with coronary artery disease (*FOXC1*), myocardial infarction (*USP39*), and atrial fibrillation (*SLC35F1* and *SSPN*) through their effects on RHR have been associated with these cardiovascular diseases in recent studies[32–35].

To obtain further biological insights into RHR, we performed pathway analyses using DEPICT and found numerous newly associated pathways. The strongest associated clusters were identified previously and their importance to RHR biology was therefore validated in the current study[7]. Conditional analyses on the tissue enrichment demonstrate that genes influencing RHR are more likely co-expressed than primarily or solely located within non-cardiovascular tissues. However, it should be noted that conditional analyses inherently attenuate tissue enrichment considering DEPICT is based on the co-regulation of gene expression[36]. Using cardiac single-nucleus RNA data, we demonstrate that RHR genes are mostly expressed in cardiomyocytes. We provide electrophysiological insights into the biology of the RHR-associated variants and show that they exert diverse effects on ECG morphology with the largest effect on ventricular repolarization.

In-depth analyses were performed to assess genetic associations of RHR with clinical outcomes. In contrast to previous observational[1,2] and MR studies[7], we do not find evidence for an association between genetically predicted RHR and all-cause mortality. Moreover, genetically predicted RHR was not associated with parental longevity nor with any of the 35 leading causes of mortality. Lack of such associations suggests that follow-up length or large heterogenetic effects of RHR on different causes of mortality are unlikely causes for the absence of an association between genetically predicted RHR and all-cause mortality[37]. We demonstrate that the most likely cause of the discrepancy between current and previous results arises from false positive findings in previous one-sample MR analyses that were caused by weak-instrument bias at lower P-value thresholds[38]. We hypothesize that RHR is not on the causal pathway to mortality itself and that previous observational studies are more likely to reflect confounders, such as stress and socio-economic status, or reversed causation, in which an individual's disease status increases both RHR and mortality risk[39,40].

The linear MR between RHR and atrial fibrillation provided suggestive evidence for an inverse relationship between RHR and atrial fibrillation, in line with a previous linear MR study[9]. We do find a significant negative exponential dose-response curve between RHR and atrial fibrillation in support of an inverse relationship, and take the non-linear MR forward as the main result considering the fractional polynomial test indicated that a non-linear model fitted the localized average causal effect estimates better than the linear model. Previous observational studies on the relationship between RHR and atrial fibrillation have shown conflicting results and have described various

relationships including inverse linear[41–43], U-shaped[43], and J-shaped[44] associations. All these association patterns support the hypothesis that individuals with a low RHR might exhibit a higher risk of atrial fibrillation development compared to those with an average RHR. A recent stratified Mendelian randomization showed an inverse genetic relationship between RHR and atrial fibrillation in individuals with an RHR below 65 bpm as well[45]. Possible mechanisms that could underly an increased risk of atrial fibrillation in individuals with a low RHR include increased left atrial stroke volume and consequent atrial remodeling due to myocyte stretching[46], or an increased vagal tone promoting global disorganization in the left atrium due to increased heterogeneity of the refractory period[47]. In contrast to the often hypothesized U-shaped or J-shaped association[43,44], we find a decreasing risk of atrial fibrillation development in those with a high RHR. One potential explanation is that previous observational studies were affected by collider bias through confounding factors which increase atrial fibrillation risk and typically occur in tandem with a high rather than a low RHR, such as hypertension[48] and obesity[49]. We advocate for a cautious interpretation of current results due to the diverse biological mechanisms through which the RHR-associated genetic variants alter the risk of atrial fibrillation development[8].

We found that genetically predicted RHR was inversely associated with risk of any, ischemic and cardio-embolic stroke. The results were not replicated in the UK Biobank, possibly due to the substantially lower amount of cases. The inverse association is in contrast to many observational studies and we therefore performed multivariable MR analyses to pinpoint biological mechanisms that could underly the discrepancy[4,5]. We showed that atrial fibrillation attenuates the protective effect of higher genetically predicted RHR on developing cardio-embolic stroke. This indicates either biological or mediated pleiotropic effects of atrial fibrillation in the association between genetically determined RHR and cardio-embolic stroke, which cannot be distinguished based on the current results. The relationship between a low RHR and cardioembolic stroke was attenuated by only 18.4% when corrected for atrial fibrillation, which might underestimated due to atrial fibrillation being a commonly missed diagnosis. Correction for atrial fibrillation only minimally affected the association between RHR and any or ischemic stroke, despite cardio-embolic stroke accounting for a substantial amount of ischemic stroke cases[50]. Although hypertension is another important risk factor for stroke, it commonly occurs in tandem with a higher and not a lower RHR[4,51,52] and we found that neither systolic nor diastolic blood pressure affects the association between RHR and stroke. We did find that pulse pressure attenuates the association between RHR and any, ischemic and large-artery stroke. Lower RHR has previously been demonstrated to increase pulse pressure due to a higher likelihood of pressure wave reflections during prolonged systole[53] and increased pulse pressure has been established as a risk factor of stroke[53–55]. Moreover, the Conduit Artery Functional Endpoint Study (CAFE) study postulated that pulse pressure underlies the inferiority of β-blocker based treatment (which lowers RHR) to amlodipine-based treatment in the prevention of stroke, despite equal effects on peripheral blood pressure[53,56]. Our results could be considered as support for this mechanism in the scenario that the RHR-associated genetic variants only affect pulse pressure through RHR and the association with stroke is primarily driven by RHR. However, MR-Steiger sensitivity analysis indicated that the association between the RHR-associated genetic variants and pulse pressure is unlikely mediated through RHR entirely. Biological pleiotropic effects are therefore more likely to cause the attenuation of the association between RHR and stroke when correcting for pulse pressure.

Finally, our study provides evidence that higher genetically predicted RHR increases the risk of developing dilated cardiomyopathy. The importance of decreasing RHR in the treatment of heart failure with reduced ejection fraction, the clinical phenotype of dilated cardiomyopathy, has been thoroughly studied. Beta-blockers have been shown to reduce mortality in individuals with heart failure with reduced ejection fraction and form the cornerstone of pharmacological treatment[57–59]. There is also evidence that ivabradine lowers cardiovascular mortality in heart failure with a reduced ejection fraction[60]. This protective effect is more likely due to its effect on RHR than heart rate variability as it has a larger effect on RHR[61]. The fact that the MR results were robust to the exclusion of SNPs associated with the −18 ms point of the R-peak, an established biomarker of dilated cardiomyopathy, supports the interpretation that current findings are driven by RHR differences that mimic pharmacological rate control[14]. Our MR on the compound definition of heart failure could be hampered by its phenotypical heterogeneity, as we were unable to differentiate between heart failure with reduced and preserved ejection fraction. It would be interesting to repeat the current MR analyses if more in-depth phenotyping on left ventricular ejection fraction and function becomes available, especially considering the different effects of RHR on familial dilated versus hypertrophic cardiomyopathy in the current study.

Several limitations should be considered. Although the current 493 RHR-associated genetic variants explained more than double the RHR variance compared to the 64 loci from our previous study[7], there is still a large gap with heritability estimates from twin studies that range between 23% and 70%[62–64]. Future studies could include whole exome sequencing data to further increase our insights into the genetic architecture of RHR[65]. Second, individuals with cardiovascular diseases were included in the GWAS, which could potentially affect exposure-outcome associations. However, post-hoc analysis showed that UK Biobank participants with a history of cardiovascular disease or who used RHR-altering medication can be jointly analyzed with participants without such a medical profile. In addition, a two-sample MR strategy was adopted, reducing the risk that potential weak-instrument bias increases type 1 error rates through the reintroduction of confounding, population stratification, or correlated pleiotropy[38]. We note the broad biological nature of RHR genetic variants as illustrated by the diverse ECG patterns the genetic variants elicit on the full cardiac cycle. These broad effects should be taken into consideration for the correct interpretation of the MR results, as pleiotropy and reversed causation might be introduced in the MR. For example, some genetic variants were included in the MR analyses which could be more specific for another trait (i.e. rs2234962 near *BAG3* for dilated cardiomyopathy). We believe that the influence of reversed causation on current results is minimal because we excluded variants more strongly associated with the outcome. The MR results were generally consistent across a multitude of sensitivity analyses, strengthening the interpretation of a true relationship. However, our study is not interventional in design, and a conservative interpretation of the results as generally unconfounded rather than causal estimates should be preferred. We stress that any causal claims can only be made if interventions or drugs alter RHR equal to the biological mechanisms in which RHR-associated genetic variants affect RHR.

In conclusion, our GWAS meta-analysis identified 493 RHR-associated genetic variants within 352 loci, to which we prioritized 670 candidate causal genes. We demonstrated cardiovascular tissues as the primary enrichment sites for RHR gene effects and showed that their gene expression is highest in cardiomyocytes. ECG signatures showed that RHR-associated genetic variants exert the largest effect on RHR through ventricular repolarization. We found no evidence for linear and non-linear associations between genetically predicted RHR and all-cause mortality across several analyses, suggesting that the well-known link between higher RHR and all-cause mortality reflects confounding factors and reversed causation. The results point towards an inverse association between genetically predicted RHR and the development of atrial fibrillation and any stroke, ischemic stroke, and cardio-embolic stroke, whereas it is positively associated with dilated

cardiomyopathy development. Multivariable MR analyses showed that atrial fibrillation attenuates the protective effect of higher RHR on the development of cardio-embolic stroke. Pulse pressure attenuates the protective effects on any stroke, ischemic and large artery stroke, but this likely reflects biological pleiotropy rather than true mediation.

## Methods

### Method details

**Populations.** The full RHR meta-analysis included 100 studies with data on RHR in up to 835,465 individuals. RHR was obtained from ECG in 54 studies, from pulse rate in 31 studies (of which seven were self-measured by the participants), from blood pressure monitor in nine studies, from electronic medical records in three studies, from manual measurement in one study and through a combination of multiple of the before mentioned methods in two studies. Further information on cohort characteristics is provided in Supplementary Data 1 and statistical details are provided in the "Genome-wide association studies" section.

**Imputation and quality control.** Genotyping and quality control before imputation were performed using different genome-wide genotyping arrays and methods, as further detailed in Supplementary Data 1.

The UK Biobank was imputed to the Haplotype Reference v1.1 panel (HRC) by the Wellcome Trust Centre for Human Genetics. Analysis has been restricted to variants that are in the HRC v1.1. Quality control of samples and variants and imputation was performed by the Wellcome Trust Centre for Human Genetics, as described in more detail elsewhere[66].

The 99 cohorts of the IC-RHR were imputed to 1000 Genomes Phase 1 and 3. For further information, please see Supplementary Data 1.

On the cohort level, we performed quality control by (1) re-formatting and SNP-name harmonization; (2) checking the used reference panel by plotting effect allele frequency plots using 1000G as a reference; (3) checking for genomic inflation by plotting QQ plots; (4) checking the betas by plotting histograms of the beta, frequency and info; (5) comparison of the expected $P$-value based on beta and standard error versus reported $P$-values.

**Association with other traits.** Genetic correlation analyses with GWAS of previously investigated traits were performed using LD Hub platform[67]. Genetic correlations were considered significant if they achieved a Bonferroni-corrected significance threshold of $P < 0.05/855 = 5.85 \times 10^{-5}$. The GWAS Catalog was queried to find previously established genetic variants ($P < 1 \times 10^{-5}$) in LD ($R^2 > 0.8$) with all 493 RHR variants[68]. Summary statistics were downloaded from the NHGRI-EBI GWAS Catalog on 27/04/2020.

**Functional annotation of genes.** For all independent genetic variants that were genome-wide significantly associated in the final meta-analysis, candidate causal genes were prioritized as followed: (1) by proximity, the nearest gene and any other gene within 10 kb; (2) protein-coding genes containing variants in LD with RHR-associated variants at $R^2 > 0.8$; (3) eQTL genes in LD with RHR-associated variants at $R^2 > 0.8$; and (4) DEPICT gene mapping using variants that achieved $P < 1 \times 10^{-8}$ (further information described below). Annotation of all identified genes was performed by querying GeneALacart[69].

**Query of dbNSFP.** The dbNSFP database (version 3a) was queried to obtain functional prediction and annotation of all potential non-synonymous genetic variants[70]. The dbNSFP database contains information on multiple prediction algorithms and conservation scores further detailed elsewhere[70].

**eQTL analyses.** Colocalization of multiple expression quantitative trait loci (eQTL) was performed using SMR and HEIDI analyses (version 0.710)[13] in data repositories from GTEx V7[71], GTEx brain[71], Brain-eMeta eQTL[72], and blood eQTL from Westra[73] and CAGE[74]. Colocalization analyses were performed to test whether the effect size of the RHR-associated variants on the phenotype is most likely mediated by gene expression[13]. eQTL genes were considered as candidate causal genes if they achieved a significance after Bonferroni correction for the amount of eQTLs tested ($P < 0.05/188,737 = 2.65 \times 10^{-7}$), passed the HEIDI test at $P > 0.05$, and if the lead variants of the eQTL genes were in LD ($R^2 > 0.8$) with the RHR-associated genetic variants.

**DEPICT analyses.** DEPICT was used to find genes associated with identified variants, enriched gene sets, and tissues in which these genes are highly expressed. DEPICT.v1.beta version rel137 (obtained from https://data.broadinstitute.org/mpg/depict/) was used to perform integrated gene function analyses as stated above[36]. DEPICT was run using all genetic variants that achieved $P < 1 \times 10^{-8}$. Genes were considered possible candite genes if FDR < 0.05, taking into account multiple hypothesis testing.

**Pathway analyses.** DEPICT was used to find enriched gene sets using the settings described above[36]. Enriched genesets were further clustered on the basis of the correlation between scores for all genes using an Affinity Propagation method as provided by DEPICT[36]. Each cluster was named according to the name of the most central gene set as identified using the Affinity Propagation method. Identified meta-clusters were compared to the clusters found in the study of Eppinga et al. and were determined to be new if not a single cluster within the meta-cluster had been identified before. Clustering was performed using DEPICT software in python 2.7[75] and visualization using Cytoscape 3.8.0[76].

**Tissue enrichment.** DEPICT was used to find enriched tissues using the settings described above[36]. Enriched tissues were further investigated by performing conditional analyses to provide evidence for an independent association with RHR. The following formula was used:

$$T_{cond} = \frac{Z_t - \rho_{ts}Z_s}{\sqrt{1 - \rho_{ts}^2}} \tag{1}$$

Here, $Z_t$ is the maximum $Z$-value of all tissue $Z$-values, $Z_s$ a vector of all tissue $Z$-values, and $\rho_{ts}$ the correlation between the tissue of $Z_t$ and the tissue of $Z_s$[77]. The maximum $Z$-value ($Z_t$) was determined for every new iteration. Conditional analyses were performed up until the highest $Z$-value reached 2.585, which corresponds to the lowest $Z$-value with an FDR < 0.05.

**ECG morphology.** The ECGenetics browser was used to gain insights into the electrophysiological effect of the RHR-associated genetic variants[14]. Detailed information on the methodology can be found in the study of Verweij et al. and is briefly discussed below. The ECGenitics browser contains genome-wide summary statistics of the complete cardiac cycle. The complete cardiac cycle was defined using two methods, including (a) the signal-averaged electrocardiographic beat surrounding the R wave at a resolution of 500 Hz resulting in 500 averaged data points and (b) R−R intervals corrected signal (made of equal length of 500 data points).

All RHR-associated genetic variants were tested for their association with both the non-normalized and normalized association patterns. A heatmap was constructed containing all associated genetic variants associated with at least one point on the ECG at a Bonferonni-corrected $P$-value of $0.05/493/(500 \times 2) = 1 \times 10^{-7}$. Effects were aligned to the most positively associated allele across all time points. The heatmap shows a hue ranging from red (positive effect) to a blue color (negative effect) color scale, with yellow indicating no effect.

Secondly, the total effect of the 493 RHR-associated genetic variants on ECG morphology was assessed using an ECG-wide MR approach (inverse variance weighted fixed-effects model) on the non-normalized and normalized association pattern.

**Single-nucleus RNA expression.** All genes prioritized in the current study were queried in the single-cell data from the study of Tucker et al. through the Broad Institute's Single Cell Portal (available at: https://singlecell.broadinstitute.org/single_cell/study/SCP498/ transcriptional-and-cellular-diversity-of-the-human-heart under study ID SCP49) to gain insights in their transcriptional and cellular diversity[15]. We selected the 86 genetic variants strongly associated with at least one ECG time point. We took forward the most likely candidate gene per genetic variant, based on the amount of gene identification strategies by which the gene was identified. When a genetic variant highlighted multiple genes identified by the same amount of gene identification strategies, we took forward the gene with the highest biological plausibility of involvement in RHR biology. A dotted heatmap was constructed for this subset of genes.

**UK Biobank definitions.** In the UK Biobank, we captured the prevalence and incidence of functional outcomes through data collected at the Assessment Centre in-patient Health Episode Statistics (HES) and data on the cause of death from the National Health Service (NHS) Information Centre. Prevalence of disease was also based on an interview with a trained nurse at the baseline visit (self-reported). HES data were available up to 31-03-2017 for English participants, 29-02-2016 for Welsh participants, and 31-10-2016 for Scottish. Information on cause of death was available for participants from England and Wales until 31-01-2018, and from the NHS Central Register Scotland for participants from Scotland until 30-11-2016. Definitions of all-cause mortality, 35 leading causes of mortality (defined as any cause of mortality with a prevalence >0.1%), coronary artery disease, myocardial infarction, atrial fibrillation, stroke (any stroke, any ischemic stroke), heart failure and subtypes (hypertrophic cardiomyopathy, dilated cardiomyopathy) are provided in Supplementary Data 15. Longevity was obtained through questionnaires in which participants were asked to provide the age of death of both parents. Individuals were excluded in case the answer was older than 115 years, if they reported themselves as adopted, if their parent was still alive but not yet long-lived, or if their parent died prematurely (fathers <46 years or mothers <57 years), in line with previously established methods[37]. Combined parental longevity was assessed by summing $Z$-scores of the age of death from both parents if the information on both parents was provided[37]. Systolic and diastolic blood pressure values were obtained through two automated and/or two manual blood pressure measurements. The average value of all available blood pressure measurements was used per phenotype. Automated measurements were corrected according to the previously described methodology[78]. In addition, we corrected systolic and diastolic blood pressure for medication use, by adding respectively 15 and 10 mmHg to the blood pressure trait[79]. Pulse pressure was calculated by subtracting diastolic from systolic blood pressure.

Statistical details of the analyses on functional outcomes are provided in the "Genetics and regression analyses on functional outcomes in the UK Biobank" and "Mendelian randomization" sections.

**External cohort definitions.** External cohorts included the CARDIoGRAMplusC4D[80], AFGen[81], MEGASTROKE[50], and ICBP-plus[19] consortia and descriptions have been detailed previously. Effect sizes from the ICBP-plus consortium were obtained from the meta-analysis of the UK Biobank and ICBP-plus, after subtracting the effects from the UK Biobank (for further details, see the section "Meta-subtract of blood pressure traits"). An overview of these studies is provided in Supplementary Data 16. We searched for proxies (LD > 0.8) in case variants could not be found within the outcome datasets.

## Quantification and statistical analysis

**Genome-wide association studies.** All included cohorts performed genetic variant association analyses on RHR using linear regression analyses assuming an additive genetic model (Supplementary Data 1). No transformation of heart rate was performed and extreme (>4 SD) phenotypic outliers were excluded analogous to previous GWAS on RHR and as per predefined criteria[7]. The GWAS model was adjusted for age, age[2], body mass index, sex, and study-specific covariates (e.g. principal components, genotyping array, and RHR measuring method in case multiple RHR methods were used within a study).

The UK Biobank GWAS was performed using BOLT-LMM v2.3beta2, employing a mixed linear model that corrects for population structure and cryptic relatedness[82]. A total of 484,307 participants from the UK Biobank remained available for the GWAS after the exclusion of 14,242 individuals for whom no genetic data was available, 1341 individuals who failed genetic quality control, 1058 individuals who were outside the 4SD range for RHR, and 1587 individuals due to missing covariates (Supplementary Data 1).

Study-specific details and methodology of the 99 cohorts of the IC-RHR are provided in Supplementary Data 1. Several software programs were used for the analysis, including mach2qtl[83], minimac[84], minimac2[85], IMPUTE2[86], GEEPACK[87], ProbaABEL[88], and MMAP[89], for which version details per cohort are provided in Supplementary Data 1. A fixed effect meta-analysis using the inverse variance method in METAL was performed on all 99 cohorts, including up to a total of 351,158 individuals[90]. Genomic control was applied at the study level by correcting for the study-specific lambda.

All genetic variants were excluded if they had poor imputation quality score (Info < 0.3) and effective sample size ($N_{\text{eff}}$) < 25 for the genetic variants computed as sample size × Info × 2 × minor allele frequency (MAF)×(1−MAF). After these exclusions, a total of 19,400,415 and 27,082,649 variants remained available for the UK Biobank and IC-RHR GWAS respectively.

Again, a fixed-effects meta-analysis using the inverse variance method in METAL was performed to pool the data from the UK Biobank and IC-RHR up to 835,465 participants using ~30M genetic variants[90]. LD score regression software (v1.0.0) was used to calculate linkage disequilibrium score regression intercepts and attenuation ratios[11,12]. We corrected for genomic inflation prior to the meta-analysis by multiplying the standard errors with the square root of linkage disequilibrium score regression intercepts in the UK Biobank (1.132 ± 0.017) and the IC-RHR (1.020 ± 0.010)[11,12].

PLINK (version 1.9) was used to prune genetic variants in a set of independently associated variants[91]. An independent genetic variant was defined as a genome-wide significant genetic variant in low LD ($R^2 < 0.005$) with another genome-wide significant variant within a five-megabase window. A genetic locus was defined as the most significant variant in a megabase region at either side of the independent genetic variant.

A single one-stage replication analysis was performed next. The following criteria had to be satisfied for a signal to be reported as a replicated signal for RHR:

1. the sentinel genetic variant has $P < 1 \times 10^{-8}$ in the discovery (UKB + IC-RHR) meta-analysis;
2. the sentinel genetic variant shows support ($P < 0.01$) in the UKB GWAS alone;
3. the sentinel genetic variant shows support ($P < 0.01$) in the IC-RHR meta-analysis alone;
4. the sentinel genetic variant has the concordant direction of effect between UKB and IC-RHR datasets;

The sentinel genetic variants were compared with previous loci from previous GWAS of RHR and were determined novel if located outside a 1-megabase distance of previously RHR-associated loci[7,8,10]. We selected the $P$-value thresholds to be an order of magnitude more

stringent than a genome-wide significance *P*-value to ensure robust results and to minimize false positive findings.

**Post-hoc quality control.** We performed additional analyses to investigate whether individuals with a history of cardiovascular disease or those who took RHR-altering medication could influence the results of the GWAS. The UK Biobank population was stratified by a medical history of any cardiovascular disease or reported intake of RHR-altering medication. A history of any cardiovascular disease was defined according to the definition in Supplementary Data 15. RHR altering medication was defined as intake of beta-blockers, calcium antagonists, sotalol, amiodarone, flecainide, anti-depressants, atropine, other anti-cholinergic medication, cardiac glycosides, diuretics, ACE-inhibitors or angiotensin II receptor blockers, analogous to previous methods[92]. Linear regressions on RHR were performed in both populations, using cluster-robust standard errors with genetic family IDs as clusters to account for the relatedness among participants. Exclusions and covariates were similar to those used for the GWAS. Individuals belonging together based on 3rd-degree or closer as indicated by the kinship matrix (kinship coefficient > 0.0442) provided by UK Biobank received a family ID. A Chow test was used to investigate whether there were significant differences in beta estimates in participants with and without cardiovascular disease or RHR-altering medication[93]. The post-hoc quality control was performed using the statistical software STATA 15 (StataCorp LP).

**Genetics and regression analyses.** All of the outcomes assessed in the UK Biobank that are reported in this manuscript have been adjusted for age, sex, the first 30 principal components (PCs) to account for population stratification, and genotyping array (Affymetrix UK Biobank Axiom® array or Affymetrix UK BiLEVE Axiom array). The exclusions were performed according to the above-mentioned methods for the GWAS of RHR in the UK Biobank. In addition, we excluded 74,471 individuals based on familial relatedness, after which 412,481 individuals remained available for further analyses.

SNP-outcome associations for all outcomes (see the section "UK Biobank definitions") were obtained for all 493 variants with a $P < 1 \times 10^{-8}$ in the RHR GWAS (see section "Mendelian randomization analyses"). The associations with all-cause mortality and 35 leading causes of mortality within the UK Biobank (defined as a prevalence higher than 0.1%) were obtained using a Cox proportional hazard model during a median (interquartile range) follow-up of 8.9 (8.2–9.5) years. The associations with parental longevity was assessed using linear regression analyses. The associations with both prevalent and incidence of cardiovascular diseases were assessed using logistic regression analyses. Cox and linear regressions were corrected for age at baseline, while the logistic regression analysis was corrected for age until the last date of follow-up to correctly account for both prevalent and incident disease.

We performed an in-depth assessment of the association between RHR and all-cause mortality in the UK Biobank by systematically altering the differences between the current study and the previous study from Eppinga et al., which included (a) the set of SNPs, (b) the *P*-value threshold for SNP inclusion, (c) the assessment of the outcome in an independent cohort, and (d) the follow-up length[7]. Genetic risk scores for RHR were created following an additive model by summing the number of alleles (0, 1, or 2) for each individual after multiplication with the effect size for RHR. Genetic risk scores were constructed using the 493 discovered variants within the full meta-analyses using the effect sizes of the IC-RHR, the effect sizes of the UK Biobank, and using the 73 previously discovered variants at five *P*-value thresholds ($1 \times 10^{-8}$, $5 \times 10^{-8}$, $1 \times 10^{-7}$, $1 \times 10^{-6}$, $1 \times 10^{-5}$). These were transformed to translate to a change of 5 bpm. The association with all-cause mortality was tested using Cox regression analyses in different populations of the UK Biobank. One population included all

individuals (ncases = 16,289, ncontrols = 396,183). Another population included a subset of individuals which were genotyped for the UK Biobank interim release from May 2015, which included in the GWAS by Eppinga et al. (ncases = 4953, ncontrols = 133,102). The final population consisted of a subset of individuals without genetic information at the time of the UK Biobank interim release, which was therefore not included in the GWAS by Eppinga et al., ncases = 11,336, ncontrols = 283,081). Please note that sample sizes might slightly differ from those in the previous GWAS due to updated exclusions. Lastly, point d) was taken into account by re-performing above-mentioned steps using mortality data up until the previously available follow-up (All individuals, ncases = 7099, ncontrols = 405,373; individuals not included in the GWAS by Eppinga et al., ncases = 5000, ncontrols = 289,417; and those included in the GWAS by Eppinga et al., ncases = 2099, ncontrols = 115,956). All regression analyses were performed using the statistical software STATA 15 (StataCorp LP).

**Mendelian randomization analyses.** All 493 independent genetic variants at $P < 1 \times 10^{-8}$ in the final meta-analysis were taken forward in the MR. To minimize overlap between exposure and outcome cohorts, effect sizes were taken from the IC-RHR data to test the associations with outcomes within the UK Biobank, whereas effect sizes were taken from the UK Biobank to test the association within other independent cohorts. Proxies (LD > 0.8) were searched in case genetic variants could not be found within the UK Biobank or IC-RHR. All effect sizes were transformed to translate to a change in RHR of 5 bpm.

Potential weak instrument bias was assessed by calculating the *F*-statistic using the following equation:

$$F = \frac{R^2(n-2)}{1-R^2} \quad (2)$$

In this formula, *n* is the sample size of the exposure and $R^2$ is the amount of variance of the exposure explained by the SNP[94]. $R^2$ was calculated based on summary statistics using a previously established formula[95]. Genetic variants were not excluded from further analyses if the *F*-statistic was <10 as this can exacerbate bias by increasing the chance of winner's curse[38]. Exposure and outcome summary statistics were then harmonized using the TwoSample MR package[96]. Forward strand alleles were inferred using allele frequency information and palindromic SNPs were removed if the MAF was above the recommended setting of 0.42[96]. MR-Steiger filtering was applied to explore pleiotropic effects through the assessment of potential reversed causation. $R^2$ for both the exposure and outcome were calculated and variants were removed from further analyses if the $R^2$ of the exposure is significantly lower (*P*-value < 0.05) than the $R^2$ of the tested outcome[97]. $R^2$ for linear traits was calculated as described above[95], $R^2$ for binary outcomes was calculated on the liability scale[98]. A true causal direction was assumed if the $R^2$ for binary outcomes was too small to be correctly estimated. Variants were excluded from further analyses in case a false causal direction was indicated.

The linear association between genetically determined RHR on all outcomes was initially assessed using the IVW multiplicative random-effects method, which provides a consistent estimate under the assumption of balanced pleiotropy. The Rücker framework was applied to assess heterogeneity and thus potential pleiotropy within the MR effect estimates[99]. A Cochran's *Q* *P*-value of <0.05 was considered as proof of heterogeneity within the IVW estimate and, as a consequence, balanced horizontal pleiotropy. An $I^2$ index >25% supports this conclusion[100]. The MR-Egger test was performed to allow SNPs to exert unbalanced horizontal pleiotropy[101]. The Rücker framework assesses heterogeneity within the MR-Egger regression (Rucker's *Q*) and calculates the difference between heterogeneity within the IVW effect estimate (*Q*–*Q*′)[99]. A significant *Q*–*Q*′ (*P* < 0.05) in combination with a significant non-zero intercept of the MR-Egger

regression ($P < 0.05$) was considered an indication of unbalanced horizontal pleiotropy. We then moved from an IVW model to the MR-Egger model as initial analysis, as the MR-Egger can provide causal estimates if SNPs exert unbalanced horizontal pleiotropy under the assumption that the Instrument Strength Independent of Direct Effect (InSIDE) assumption holds. Weak instrument bias in the MR-Egger regression analysis was assessed by $I^2_{GX}$ and was considered to indicate a low risk of measurement error if larger than 95%[102]. The MR-Lasso method was used to find consistent estimates under the same assumptions as the IVW method, but only for the set of genetic variants not identified as outlier[103]. This method is most valuable in the scenario that a small proportion of the genetic variants is invalid and shows heterogeneous ratio estimates[103]. The weighed median approach was used to provide a consistent estimate if up to half of the variants are invalid. Finally, we performed the MR contamination mixture method to provide a consistent estimate if no larger subset of invalid genetic variants estimates the same causal association than the subset of valid genetic variants[104].

The non-linear associations of genetically predicted RHR with all-cause mortality and cardiovascular diseases were assessed using a fractional polynomial method[105,106]. These associations were assessed using the UK Biobank as an outcome cohort, considering this was the largest cohort with individual-level data available to us. Consequently, we used the independent weights of the IC-RHR meta-analysis to construct a weighted polygenetic risk score of RHR by summing the number of alleles (0, 1, or 2) for each individual after multiplication with the effect size between the genetic variant and RHR. We first calculate residual RHR by subtracting the results of the regression of RHR from the polygenetic score of RHR from RHR itself. Covariates of the regression included age, age$^2$, sex, BMI, genotyping array, and PC1-PC30, analogous to the GWAS. Residual RHR, which characterizes the predicted RHR for an individual if their polygenetic score took the value zero, was then divided into 30 quantiles. Stratifying on residual RHR rather than total RHR avoids overadjustment and collider bias as residual RHR is not downstream of the effect of the genetic variants on the outcome in a causal diagram. We then calculated the genetic associations with the exposure in each stratum of residual RHR using linear regression analyses, correcting for the same covariates as described above. Two tests for non-linearity in the genetic association with the exposure (trend and Cochran's $Q$ tests) were performed to investigate heterogeneity in the polygenetic score of RHR on residual RHR in different strata. We then calculated the genetic associations with the outcome in each stratum. The same methodology was used as described in the section "Genetics and regression analyses", including the same covariate model (age, sex, genotyping array, and PC1-PC30) and regression type (Cox regression for all-cause mortality and logistic regression for cardiovascular diseases). The outcome regression coefficient was then divided by the exposure regression coefficient as a ratio of coefficients to obtain local average causal effects (LACE) in each stratum. These localized average causal effects were meta-regressed against the mean of the exposure in each stratum in a flexible semiparametric framework, using the derivative of fractional polynomial models of degrees 1 and 2. All possible fractional polynomials of degrees 1 and 2 were fitted using the powers −2, −1, 0, 0.5, 1, 2, and 3[106]. The fractional polynomial of degree 1 is fit to the data if the fractional polynomial of degree 1 was as good of a fit ($P > 0.05$) as the degree 2 as indicated by the likelihood ratio test. Three tests for non-linearity of the association between genetically predicted RHR and the outcomes are reported: a trend test, which assesses for a linear trend among the localized average causal effect estimates, a Cochran's $Q$ test, and a fractional polynomial test, which assesses whether a non-linear model fits the localized average causal effect estimates better than a linear model. Please note that before fitting the fractional polynomials, we subtracted 45 from the values of RHR as the most flexible fit is achieved when the exposure is close to 0 but still positive.

A reference of RHR of 70 bpm was taken as this was close to the mean RHR of 69.3 bpm. An additional 1390 individuals were dropped compared to the linear MR estimates obtained from the UK Biobank cohort due to missing BMI values necessary for the correction of the exposure regression coefficients.

A multivariable MR approach was used to gain additional insights into the relationship between RHR (effect sizes of the UK Biobank) and (subtypes of) stroke from the MEGASTROKE consortium. We used either atrial fibrillation (AFgen consortium[81]), systolic blood pressure, diastolic blood pressure, or pulse pressure (ICBP consortium, please see the section "Meta-substract of blood pressure traits" for further details[19]) as secondary exposures to obtain insights in the direct effect of RHR on (subtypes of) stroke that are independent of this secondary exposures[107]. First, a multivariable MR-IVW method was used, in which for each exposure the instruments are selected and regressed together against the outcome, weighting for the inverse variance of the outcome[107]. Weak instrument bias for any of the exposures was assessed using $Q_{x1}$ and $Q_{x2}$[107]. When both are larger than the critical value on the $\chi^2$ distribution, there is little evidence of weak instrument bias. The critical value on the $\chi^2$ distribution was calculated by subtracting one degree of freedom from the amount of SNPs at a $P$-value of 0.05. $Q_a$ was considered to indicate potential pleiotropy when larger than the critical value on the $\chi^2$ distribution as calculated by the amount of SNPs minus two degrees of freedom at a $P$-value of 0.05[107]. Multivariable MR-Egger was performed to allow for unbalanced horizontal pleiotropy[108]. An MR-Egger intercept with a $P$-value < 0.05 in combination with a significant $Q_a$ was considered proof of unbalanced horizontal pleiotropy, and consequently the MR-Egger regression to provide a robust causal estimate[108]. Multivariable MR-Lasso analysis was performed as this method provides consistent estimates even when half of the genetic variants are invalid instruments and display unbalanced pleiotropy[109]. We also performed multivariable weighted median analysis as this type of analysis has been shown to perform well under higher levels of pleiotropy[109]. We did not search for proxies in the multivariable MR setting as this could introduce uncertainty through different LD patterns between the secondary exposure and outcome and we therefore re-estimate the univariable effect of RHR on the outcome with the eligible SNPs to allow for a better comparison of the results.

We assessed whether the Wald estimates between the RHR-associated genetic variants and the cardiovascular disease outcomes could identify risk loci not previously associated with these outcomes in their respective GWASs. The genetic variants were considered associated with the outcome if (a) the Wald estimates had concordant effects within the UK Biobank as well as either the CARDIoGRAMplusC4D[80], AFGen[81], or MEGASTROKE[50] cohorts, (b) when the Wald estimates were significant at a Bonferonni corrected threshold of $P < 1.01 \times 10^{-4}$, that is, $\alpha = 0.05$ with Bonferroni correction for a maximum of 493 independent tests, and (c) the genetic variant did not reach a genome-wide significant threshold of $P$-value $< 5 \times 10^{-8}$ in either one of the outcome cohorts used in the current study.

MR analyses were performed using R (version 3.6.3), the Two-SampleMR package (version 0.5.3)[96], and the MR-Lasso source code[103]. The multivariable MR analyses were performed using the MVMR (version 0.3)[107] and MendelianRandomization (version 0.5.1) packages[110]. Non-linear MR analyses were performed based on previously described methods[105]. For the MR on (subtypes of) mortality, we considered a liberal two-sided $P$-value of $P < 0.05$ significant for any of the outcomes using the IVW-MR random effects model. For the MR on cardiovascular diseases, we considered a Bonferroni corrected two-sided $P$-value for the amount of unique outcomes ($P = 0.05/12 = 4.17 \times 10^{-3}$) to be significant for the main IVW-MR random effects analyses, and a two-sided $P$-value between $4.17 \times 10^{-3}$ and 0.05 to indicate suggestive evidence for an association. A two-sided $P$-value threshold of $P < 0.05$ was adopted for the sensitivity analyses.

**Meta-subtract of blood pressure traits.** We used the MetaSubtract (version 1.60) package in R to remove the effects sizes of the UK Biobank from the largest blood pressure GWAS's to date in order to obtain the independent effect sizes of the ICBP consortium[19,111]. Effect sizes for the UK Biobank were obtained through linear regression analyses using every RHR SNP available in the UK Biobank as exposure, and systolic, diastolic, and pulse pressure as outcomes. Covariates included age, age$^2$, sex, BMI, genchip, and PC1-PC30. Cluster-robust standard errors with genetic family IDs as clusters were used to account for the relatedness among participants. Individuals belonging together based on 3rd-degree or closer as indicated by the kinship matrix (kinship coefficient > 0.0442) provided by UK Biobank received a family ID. To keep the cohort similar to the one used in the study from Evangelou et al., we excluded those who self-reported as of non-European ancestry ($n$ = 18,405) and pregnant women ($n$ = 306) from the 484,307 included in the GWAS, leaving 465,659 individuals for the analysis[19]. We note that we did not correct the standard errors for the genomic inflation reported for the GWASs of blood pressure traits in the UK Biobank as our linear regression estimates, while resembling the GWAS data, will not be exactly equal to BOLT-LMM estimates due to different methodologies[82].

### Reporting summary

Further information on research design is available in the Nature Portfolio Reporting Summary linked to this article.

## Data availability

All data supporting the findings described in this manuscript are available in the article and its Supplementary Information files. The genome-wide summary statistics, excluding the 23andMe data, generated in this study have been deposited in a Mendeley database available through https://data.mendeley.com/datasets/9b725x7mvb/1. The top 10,000 genetic variants, including the 23 and Me data, can be downloaded from the same repository. The full GWAS summary statistics, including 23andMe data, will be made available through 23andMe to qualified researchers under an agreement with 23andMe that protects the privacy of the 23andMe participants. Datasets will be made available at no cost for academic use. Please visit https://research.23andme.com/collaborate/#dataset-access/ for more information and to apply to access the data. The estimated average review time is 3 months. Once this has been approved, applicants can send the confirmation to the lead author of the manuscript to receive the full summary statistics. The raw data of all cohorts are protected and are not available due to data privacy laws. Referenced datasets can be obtained through their respective publications cited in the manuscript, or otherwise be accessed by the URLs provided below. Referenced data includes databases from dbNSFP (https://sites.google.com/site/jpopgen/dbNSFP), public eQTL repositories (https://cnsgenomics.com/software/smr/#DataResource), GWAS catalog (https://www.ebi.ac.uk/gwas/home), GeneALacart (https://genealacart.genecards.org/), LD hub (http://ldsc.broadinstitute.org/), ECGenetics (http://www.ecgenetics.org), single nucleus RNA expression (https://singlecell.broadinstitute.org/single_cell/study/SCP498/transcriptional-and-cellular-diversity-of-the-human-heart#study-summary). Publicly available GWAS summary statistics were used for the Mendelian randomization analyses, further information and URLs are detailed in Supplementary Data 16.

## Code availability

The analysis in the current manuscript was performed using previously published software and code. Further information on scripts and coding required to reproduce this work is available from the Lead Contact upon request.

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

## Acknowledgements

We thank all participants for their participation and valuable contributions. This research has been conducted using the UK Biobank Resource under application number 12010. The work of N.V. was supported by NWO VENI grant 016.186.125. We thank 23andMe and the 23andMe Research Team for their contribution sharing their data and performing the GWAS analysis in the 23andMe cohort. P.V. received an unrestricted grant from GlaxoSmithkline to build the CoLaus study. N.J.T. is a Wellcome Trust Investigator (202802/Z/16/Z), is the PI of the Avon Longitudinal Study of Parents and Children (MRC & WT 217065/Z/19/Z), is supported by the University of Bristol NIHR Biomedical Research Centre (BRC-1215-2001), the MRC Integrative Epidemiology Unit (MC_UU_00011/1) and works within the CRUK Integrative Cancer Epidemiology Program (C18281/A29019). A detailed list of acknowledgements and funding is provided in Supplementary Data 1 per cohort. We also thank all individuals that contributed to the generation of software programs, algorithms and genetic summary statistics. Support for title page creation and format was provided by AuthorArranger, a tool developed at the National Cancer Institute. Authors involved in the funding of the cohorts are listed below. *Meta-analyses, Lifelines, PREVEND, UK Biobank*: P.v.d.H.; *ADDITION-PRO*: T.H.; *ADVANCE*: C.I.; *ADVANCE*: T.A.; *AGES*: L.Launer, V.G.; *ASCOT*: P.Sever, P.B.M.; *BC1936*: N.G.; *BioMe*: E.P.B., R.J.F.L.; *BRIGHT*: P.B.M.; *CHS*: B.M.P.; *CoLaus*: P.V.; *Croatia-Korcula*: O.P., C.H.; *DCCT/EDIC*: D.R., The DCCT/EDIC Research Group, A.D.P.; *DESIR*: B.B., P.F.; *DGI*: L.G.; *EPIC-Norfolk*: N.J.W.; *ERF*: C.M.v.D.; *Fenland*: N.J.W.; *FINCAVAS*: M.K.; *Finrisk*: M.P.; *FUSION*: M.B.; *GENOA*: P.A.P., S.L.R.K.; *GerMIFS*: H.Schunkert, J.E.; *GoDARTS*: C.N.A.P; *GOOD*: M.L., C.O.; *HBCS*: J.G.E.; *HERITAGE*: T.R., D.C.R., C.B.; *HPFS / NHS*: P.K.; *HRS*: D.R.Weir.; *HYPERGENES*: K.S.S., D.C.; *InCHIANTI*: S.Bandinelli, L.Ferrucci; *INGI-CARL*: M.P.C.; *INGI-FVG*: G.G.; *JHS*: A.Correa; *KORA F3*: T.M., S.K.; *KORA S4*: K.Strauch., A.P.; *LOLIPOP*: J.C.C., J.S.K.; *LURIC*: W.M.; *MESA*: J.I.R.; *MICROS*: A.H.; *MPP*: O.M.; *J.G.S.; *NBS*: L.A.L.M.K.; *NEO*: R.d.M.; *NESDA*: B.W.J.H.P.; *NSPHS*: Å.J.; *ORCADES*: J.F.W.; *PIVUS*: L.L.; *PROSPER*: P.W.M., J.W.J.; *SardiNIA*: E.L.L.; *SCES*: C.Y.C.; *SHIP*: S.B.F., M.D.; *SIMES*: T.Y.W.; *TRAILS*: A.J.O.; *TWINS*: J.O.; *ULSAM*: A.P.M., C.Lindgren; *YFS*: O.T.R., T.L.

## Author contributions

**Conceptualization:** Y.J.v.d.V., R.N.E., M.Y.v.d.E., N.V., P.v.d.H. **Methodology, analysis and/or software:** *Meta-analyses, downstream annotation and Mendelian randomization, Lifelines, PREVEND, UK Biobank:* Y.J.v.d.V., R.N.E., M.Y.v.d.E., Y.P.H., M.A.S., N.V., P.v.D.H.; *23andMe:* 23andMe Research Team; *ADDITION-PRO:* Y.Mahendran; *ADVANCE:* E.S.; *AGES:* A.V.S.; *ALSPAC:* V.Y.T, N.J.T.; *ARIC:* D.E.A., P.S.d.V.; *ARIC:* A.C.M.; *ASCOT:* I.N., P.B.M.; *B58C:* D.P.S.; *BC1936:* E.V.A.; *BioMe:* C.Schurmann; *BRIGHT:* I.N., P.B.M.; *CHS:* J.A.B.; *CoLaus:* R.R., P.V.; *Croatia-Korcula:* O.P., C.H.; *DCCT/EDIC:* D.R., A.D.P.; *DECODE:* G.Sveinbjornsson, H.H, D.F.G, D.O.A.; *DESIR:* C.Lecoeur; *DGI:* C.Ladenvall; *EPIC-Norfolk:* J.H.Z., N.J.W.; *ERF:* A.I., C.T.S., C.M.v.D.; *FamHS:* L.W., M.F.F.; *Fenland:* J.Luan, N.J.W.; *FHS:* S.J.H, C.J.O.; *FINCAVAS:* N.M.; *Finrisk:* K.A., P.S.; *FUSION:* A.U.J.; *GENOA:* L.F.B.; *GerMIFSs:* L.Z.; *GoDARTS:* N.Shah; *GOOD:* M.N.; *HBCS:* J.Lahti; *HERITAGE:* T.R., D.C.R., C.B.; *HNR:* S.Pechlivanis; *HPFS/NHS:* L.Q., P.K., M.C.C.; *HRS:* W.Zhao, J.A.S., J.D.F.; *HYPERGENES:* F.R., D.C.; *InCHIANTI:* T.T.; *INGI-CARL:* A.R, M.P.C, S.U.; *INGI-FVG:* M.C.; *JHS:* L.Lange; *KORA F3:* M.M.N.; *KORA S4:* C.R., M.F.S.; *LOLIPOP:* W.Zhang; *LURIC:* M.E.K., G.E.D.; *MESA:* X.G., J.Y.; *MICROS:* F.P., L.Foco; *MPP:* J.G.S.; *NBS:* T.E.G.; *NEO:* R.N., R.d.M., D.O.M.K.; *NESDA:* Y.Milaneschi, E.J.C.d.G.; *NSPHS:* Å.J., U.G.; *ORCADES:* K.E.S., P.K.J.; *Other contributors (non-affiliated to a cohort):* S.Burgess; *PIVUS:* M.d.H.; *POPGEN:* F.D.; *PROSPER:* S.T.; RS: M.E.v.d.B., M.E.; *SardiNIA:* G.P., K.V.T.; *SCES:* Y.C.T.; *SHIP:* S.W.; *SIMES:* X.S.S., E.S.T.; *SINDI:* H.L.L.; D.Q.Q.; *TRAILS:* P.J.v.d.M.; *TWINS:* I.M.N.; *YFS:* L.P.L.; **Resources (study management, subject recruitment and genotyping):** *Meta-analyses, Lifelines, PREVEND, UK Biobank:* Y.J.v.d.V., R.N.E., M.Y.v.d.E., Y.P.H., N.V., P.v.d.H.; *23andMe:* 23andMe Research Team; *ADDITION-PRO:* D.R.Witte, T.H.; *ADVANCE:* C.I, T.A.; *AGES:* L.Launer, V.G.; *ALSPAC:* S.M.R., N.J.T.; *ARIC:* D.E.A.; *ASCOT:* P.Sever, P.B.M.; *B58C:* D.P.S.; *BC1936:* A.L., N.G.; *BioMe:* C.Schurmann, E.P.B., R.J.F.L.; *BRIGHT:* S.P., P.B.M.; *CHS:* B.M.P., S.R.H.; *CoLaus:* R.R., P.V.; *Croatia-Korcula:* O.P., I.K., C.H.; *DCCT/EDIC:* The DCCT/EDIC Research Group; *DECODE:* H.H., D.O.A.; *DESIR:* B.B., P.F.; *DGI:* L.G.; *EPIC-Norfolk:* J.H.Z., N.J.W.; *ERF:* A.I., C.M.v.D.; *FamHS:* L.W., M.F.F.; *Fenland:* J.Luan; N.J.W.; *FHS:* S.J.H., C.H.N.C., C.J.O.; *FINCAVAS:* N.M., K.N., M.K.; *Finrisk:* M.P.; *FUSION:* K.L.M., M.B.; *GENOA:* P.A.P., S.L.R.K.; *GerMIFSs:* H.Schunkert, J.E.; *GoDARTS:* C.N.A.P; *GOOD:* M.L., C.O.; *GS:SFHS:* A.Campbell, S.Padmanabhan, D.J.P.; *HBCS:* J.Lahti., J.G.E.; *HERITAGE:* T.R., D.C.R., C.B.; *HNR:* S.M.; *HPFS / NHS:* P.K., M.C.C.; *HRS:* W.Zhao, J.A.S., J.D.F., D.R.Weir.; *HYPERGENES:* F.R., K.S.S., D.C.; *InCHIANTI:* S.Bandinelli, L.Ferrucci; *INGI-CARL:* A.R., S.U.; *INGI-FVG:* M.C., G.Sinagra, G.G.; *JHS:* A.Correa; *KORA F3:* M.M.N., M.W., T.M., S.K.; *KORA S4:* K.Strauch, A.P.; *LOLIPOP:* J.C.C., J.S.K.; *LURIC:* M.E.K., W.M.; *MESA:* K.D.T., J.I.R.; *MICROS:* A.H.; *MPP:* O.M., J.G.S.; *NBS:* T.E.G., J.d.G., L.A.L.M.K.; *NEO:* R.d.M., D.O.M.K.; *NESDA:* Y.Milaneschi, B.W.J.H.P.; *NSPHS:* Å.J., U.G.; *ORCADES:* J.F.W.; *PIVUS:* L.L., J.S.; *POPGEN:* A.F., W.L.; *PROSPER:* S.T., P.W.M., J.W.J.; *SardiNIA:* E.L.L.; *SCES:* Y.C.T., N.T., C.Y.C.; *SHIP:* S.W., M.D.; *SIMES:* E.S.T., T.Y.W.; *SINDI:* D.Q.Q., C.Sabanayagam; *TRAILS:* H.Snieder, A.J.O.; *TWINS:* I.M.N., J.O., H.R.; *ULSAM:* A.P.M., C.Lindgren, M.I.; *YFS:* L.P.L., O.T.R., T.L.; **Data curation:** R.N.E., Y.J.v.d.V., Y.P.H., M.Y.v.d.E., N.V., P.v.d.H. **Writing – Original Draft:** Y.J.v.d.V., R.N.E., M.Y.v.d.E., N.V., P.v.d.H. **Interpretation of data:** *Meta-analyses, Lifelines, PREVEND, UK Biobank:* Y.J.v.d.V., R.N.E., M.Y.v.d.E., Y.P.H., N.V., P.v.d.H.; *ADDITION-PRO:* Y.Mahendran, T.H.; *ADVANCE:* E.S., T.A.; *AGES:* A.V.S.; *ALSPAC:* V.Y.T, N.J.T.; *ARIC:* D.E.A., P.S.d.V., A.C.M.; *B58C:* D.P.S.; *BC1936:* E.V.A., N.G.; *BioMe:* C.Schurmann, E.P.B., R.J.F.L.; *CHS:* J.A.B., N.Sotoodehnia, B.M.P., S.R.H.; *CoLaus:* R.R.; *Croatia-Korcula:* O.P., I.K., C.H.; *DCCT/EDIC:* D.R., A.D.P.; *DECODE:* G.Sveinbjornsson, H.H., D.F.G, D.O.A., K.Stefansson; *DGI:* C.Ladenvall; *EPIC-Norfolk:* J.H.Z., N.J.W.; *ERF:* A.I.; C.M.v.D.; *FamHS:* M.F.F.; *Fenland:* J.Luan, N.J.W.; *FHS:* S.J.H., C.J.O.; *Finrisk:* K.A. P.S., M.P.; *GENOA:* L.F.B., P.A.P.; *GerMIFSs:* H.Schunkert, J.E.; *GoDARTS:* N.Shah, C.N.A.P; *HBCS:* J.Lahti., J.G.E.; *HERITAGE:* T.R., D.C.R., C.B.; *HYPERGENES:* D.C.; *InCHIANTI:* T.T., S.Bandinelli, L.Ferrucci; *JHS:* L.Lange, A.Correa; *KORA F3:* M.M.N., M.W.; *KORA S4:* C.R., M.F.S.; *LURIC:* W.M.; *MESA:* X.G., H.J.L., K.D.T., J.I.R.; *MPP:* J.G.S.; *NESDA:* B.W.J.H.P., Å.J.; *NSPHS:* U.G.; *Other contributors (non-affiliated to a cohort):* S.Burgess; *PIVUS:* M.d.H.; *SardiNIA:* K.T.V., E.L.L.; *SHIP:* S.W.; *SIMES:* X.S.S.; *SINDI:* H.L.L.; *TRAILS:* H.Snieder; *TWINS:* I.M.N.; *ULSAM:* A.P.M., C. Lindgren. **Review & editing:** All authors reviewed and had the opportunity to comment on the manuscript. **Visualization:** Y.J.v.d.V., N.V., R.N.E., Y.P.H., M.A.S.

**Article** https://doi.org/10.1038/s41467-023-39521-2

## Competing interests

N.V. is currently employed at Regeneron plc. The 23andMe Research team members are current or former employees of 23andMe, Inc. and hold stock or stock options in 23andMe. P. Sever has received research awards from Pfizer Inc. I.N. is now a full-time employee at Gilead. B.M.P. serves on the Steering Committee of the Yale Open Data Access Project funded by Johnson & Johnson. K.S., H.H., D.F.G., D.O.A., and G.S. are employees of deCODE genetics/Amgen Inc. C.R. is currently employed at the Broad institute and is supported by a grant from Bayer AG to the Broad Institute focused on the development of therapeutics for cardiovascular disease. M.L. receives consulting or lecturing fees from Amgen, Astellas, UCB, Consilient Health, GE/Lunar, Tromp/Hologic, Renapharma, Meda/Mylan, Janssen-Cilag and Radius Health. W.M. is employed with Synlab Holding Deutschland GmbH. M.E.K. is employed with Synlab Holding Deutschland GmbH and reports grants and personal fees from AMGEN, BASF, Sanofi, Siemens Diagnostics, Aegerion Pharmaceuticals, Astrazeneca, Danone Research, Numares, Pfizer, Hoffmann LaRoche; personal fees from MSD, Alexion; grants from Abbott Diagnostics, all outside the submitted work. The remaining authors declare no competing interests.

## Additional information

Yordi J. van de Vegte [1,179], Ruben N. Eppinga[2,179], M. Yldau van der Ende[3], Yanick P. Hagemeijer [1,4], Yuvaraj Mahendran[5], Elias Salfati[6,7], Albert V. Smith [8], Vanessa Y. Tan[9,10], Dan E. Arking[11], Ioanna Ntalla[12], Emil V. Appel[5], Claudia Schurmann [13], Jennifer A. Brody[14], Rico Rueedi[15,16], Ozren Polasek[17,18], Gardar Sveinbjornsson[19], Cecile Lecoeur [20], Claes Ladenvall[21,22], Jing Hua Zhao[23], Aaron Isaacs[24], Lihua Wang[25], Jian'an Luan [26], Shih-Jen Hwang[27], Nina Mononen[28,29], Kirsi Auro[30,31], Anne U. Jackson [32], Lawrence F. Bielak[33], Linyao Zeng[34], Nabi Shah[35,36], Maria Nethander [37,38], Archie Campbell[39,40], Tuomo Rankinen[41], Sonali Pechlivanis[42], Lu Qi[43], Wei Zhao [33,44], Federica Rizzi[45], Toshiko Tanaka[46], Antonietta Robino[47], Massimiliano Cocca[47], Leslie Lange[48], Martina Müller-Nurasyid[49,50,51], Carolina Roselli [51,52], Weihua Zhang [53,54], Marcus E. Kleber [55,56], Xiuqing Guo[57,58], Henry J. Lin[57,58], Francesca Pavani[59], Tessel E. Galesloot[60], Raymond Noordam [61], Yuri Milaneschi[62], Katharina E. Schraut[63], Marcel den Hoed[64], Frauke Degenhardt[65], Stella Trompet [61,66], Marten E. van den Berg[67], Giorgio Pistis[68,69], Yih-Chung Tham[70], Stefan Weiss [71,72], Xueling S. Sim [73], Hengtong L. Li[70], Peter J. van der Most [74], Ilja M. Nolte [74], Leo-Pekka Lyytikäinen [28,29,75], M. Abdullah Said [1], Daniel R. Witte[76], Carlos Iribarren[77,78], Lenore Launer [79], Susan M. Ring [9,10], Paul S. de Vries[80], Peter Sever[81], Allan Linneberg [82,83], Erwin P. Bottinger[13,84], Sandosh Padmanabhan[85], Bruce M. Psaty [86], Nona Sotoodehnia[87], Ivana Kolcic[17,18], The DCCT/EDIC Research Group*, David O. Arnar[19,88,89], Daniel F. Gudbjartsson [19,90], Hilma Holm [19], Beverley Balkau[91,92,93], Claudia T. Silva[94], Christopher H. Newton-Cheh [95], Kjell Nikus[96,97], Perttu Salo[30,31], Karen L. Mohlke[98], Patricia A. Peyser [33], Heribert Schunkert [34,99], Mattias Lorentzon[37,100,101], Jari Lahti [102], Dabeeru C. Rao[103], Marilyn C. Cornelis[104], Jessica D. Faul[44], Jennifer A. Smith [33,44], Katarzyna Stolarz-Skrzypek[105], Stefania Bandinelli[106], Maria Pina Concas [47], Gianfranco Sinagra[107], Thomas Meitinger [108,109], Melanie Waldenberger [110,111,112], Moritz F. Sinner [112,113], Konstantin Strauch[50,51,114], Graciela E. Delgado[55], Kent D. Taylor [57,58], Jie Yao[57,58], Luisa Foco [59], Olle Melander[115,116], Jacqueline de Graaf[60], Renée de Mutsert[117], Eco J. C. de Geus[118], Åsa Johansson [119], Peter K. Joshi [120], Lars Lind[121], Andre Franke[65], Peter W. Macfarlane [122], Kirill V. Tarasov [123], Nicholas Tan[70], Stephan B. Felix[72,124], E-Shyong Tai[70,125], Debra Q. Quek[70], Harold Snieder[74], Johan Ormel [126], Martin Ingelsson[127], Cecilia Lindgren[128], Andrew P. Morris [128], Olli T. Raitakari[129,130,131], Torben Hansen[5], Themistocles Assimes [6], Vilmundur Gudnason[7,132], Nicholas J. Timpson [133,134], Alanna C. Morrison[80], Patricia B. Munroe [12,135], David P. Strachan [136], Niels Grarup [5], Ruth J. F. Loos [5,13,137], Susan R. Heckbert[138], Peter Vollenweider[139], Caroline Hayward[140], Kari Stefansson[19,88], Philippe Froguel [141,142,143], Leif Groop[22,144], Nicholas J. Wareham[26], Cornelia M. van Duijn[94], Mary F. Feitosa [25], Christopher J. O'Donnell[145], Mika Kähönen[146,147], Markus Perola[30,31], Michael Boehnke [32], Sharon L. R. Kardia[33],

Jeanette Erdmann [148], Colin N. A. Palmer [35], Claes Ohlsson [37,149], David J. Porteous [150], Johan G. Eriksson[151,152,153], Claude Bouchard [41], Susanne Moebus [42,154], Peter Kraft [155], David R. Weir [44], Daniele Cusi [45,156], Luigi Ferrucci [46], Sheila Ulivi[47], Giorgia Girotto [47,157], Adolfo Correa [158], Stefan Kääb [112,113], Annette Peters [111,112,159], John C. Chambers[53,54,160], Jaspal S. Kooner [54,81,161], Winfried März[55,162], Jerome I. Rotter [57,58], Andrew A. Hicks [59], J. Gustav Smith[163,164,165], Lambertus A. L. M. Kiemeney [60], Dennis O. Mook-Kanamori[117,166], Brenda W. J. H. Penninx[62], Ulf Gyllensten [119], James F. Wilson [120,140], Stephen Burgess [167], Johan Sundström [121], Wolfgang Lieb [168], J. Wouter Jukema [66,169,170], Mark Eijgelsheim[67,171], Edward L. M. Lakatta [123], Ching-Yu Cheng [70,172,173], Marcus Dörr [72,124], Tien-Yin Wong [70,172,174], Charumathi Sabanayagam [70,172], Albertine J. Oldehinkel [175], Harriette Riese[126], Terho Lehtimäki [28,29], Niek Verweij[1] & Pim van der Harst[1,3,176] ✉

[1]Department of Cardiology, University of Groningen, University Medical Center Groningen, Groningen 9700RB, the Netherlands. [2]Department of Cardiology, Isala Zwolle ziekenhuis, Zwolle 8025 AB, the Netherlands. [3]Department of Cardiology, University medical Center Utrecht, Utrecht 3584 Cx, the Netherlands. [4]Analytical Biochemistry, University of Groningen, Groningen 9713 AV, the Netherlands. [5]The Novo Nordisk Foundation Center for Basic Metabolic Research, Faculty of Health and Medicine Science, University of Copenhagen, Copenhagen Ø 2100, Denmark. [6]Department of Medicine, Stanford University School of Medicine, Stanford 94305, USA. [7]Faculty of Medicine, University of Iceland, Reykjavik 101, Iceland. [8]Department of Biostatistics, University of Michigan, Ann Arbor MI48109, USA. [9]Bristol Medical School, Population Health Sciences, University of Bristol, Bristol BS82BN, UK. [10]MRC Integrative Epidemiology, University of Bristol, Bristol BS82BN, UK. [11]McKusick-Nathans Institute, Department of Genetic Medicine, Johns Hopkins University School of Medicine, Baltimore 21215, USA. [12]Clinical Pharmacology and Precision Medicine, William Harvey Research Institute, Barts and The London Faculty of Medicine and Dentistry, Queen Mary University of London, London EC1M 6BQ, UK. [13]The Charles Bronfman Institute for Personalized Medicine, The Icahn School of Medicine at Mount Sinai, New York 10029, USA. [14]Medicine, University of Washington, Seattle 98195, USA. [15]Department of Computational Biology, University of Lausanne, Lausanne 1015, Switzerland. [16]Swiss Institute of Bioinformatics, Lausanne 1015, Switzerland. [17]Department of Public Health, University of Split School of Medicine, Split 21000, Croatia. [18]Algebra LAB, Algebra University College, Zagreb 10000, Croatia. [19]deCODE genetics / Amgen Inc., Reykjavik 102, Iceland. [20]UMR 8199, University of Lille Nord de France, Lille 59000, France. [21]Clinial Genomics Uppsala, Department of Immunology, Genetics and Pathology, Science for Life Laboratory, Uppsala University, Uppsala 75185, Sweden. [22]Lund University Diabetes Center, Department of Clinical Sciences, Lund University, Malmö 20502, Sweden. [23]BHF Cardiovascular Epidemiology Unit, Department of Public Health and Primary Care, Victor Phillip Dahdaleh Heart & Lung Research Institute, University of Cambridge, Cambridge CB2 0BB, UK. [24]CARIM School for Cardiovascular Diseases, Maastricht Centre for Systems Biology (MaCSBio), Department of Physiology, Maastricht University, Maastricht 6229ER, Netherlands. [25]Division of Statistical Genomics, Department of Genetics, Washington University School of Medicine, St. Louis, MO 63108-2212Campus Box 8506 USA. [26]MRC Epidemiology Unit, University of Cambridge School of Clinical Medicine, Cambridge CB2 0QQ, UK. [27]Division of Intramural Research, National Heart Lung and Blood Institute, NIH, USA, Framingham 1702, USA. [28]Department of Clinical Chemistry, Fimlab Laboratories, Tampere FI-33014, Finland. [29]Department of Clinical Chemistry, Finnish Cardiovascular Research Center - Tampere, Faculty of Medicine and Health Technology, Tampere University, Tampere FI-33014, Finland. [30]Department of Health, unit of genetics and biomarkers, , National Institute for Health and Welfare, Finland, Helsinki FI-00290, Finland. [31]Department of molecular medicine, University of Helsinki, Helsinki FI-00290, Finland. [32]Department of Biostatistics and Center for Statistical Genetics, University of Michigan, Ann Arbor, MI 48109, USA. [33]Department of Epidemiology, University of Michigan, Ann Arbor, MI 48109, USA. [34]Department of Cardiology, Deutsches Herzzentrum München, Technische Universität München, Munich 80636, Germany. [35]Division of Population Health and Genomics, School of Medicine, University of Dundee, Dundee DD1 9SY, UK. [36]Pharmacogenetics Research Lab, Department of Pharmacy, COMSATS University Islamabad, Abbottabad 22060, Pakistan. [37]Sahlgrenska Osteoporosis Centre, Institute of Medicine, Sahlgrenska Academy, University of Gothenburg, Gothenburg 41345, Sweden. [38]Bioinformatics Core Facility, Sahlgrenska Academy, University of Gothenburg, Gothenburg 40530, Sweden. [39]Centre for Genomic and Experimental Medicine, Institute of Genetics & Cancer, University of Edinburgh, Edinburgh EH4 2XU, UK. [40]Usher Institute for Population Health Sciences and Informatics, The University of Edinburgh, Edinburgh EH16 4UX, UK. [41]Human Genomics Laboratory, Pennington Biomedical Research Center, Baton Rouge, LA 70808, USA. [42]Institute for Medical Informatics, Biometry and Epidemiology, University Hospital of Essen, University Duisburg-Essen, Essen 45122, Germany. [43]Department of Epidemiology, Tulane University, New Orleans, LA 70112, USA. [44]Survey Research Center, Institute for Social Research, University of Michigan, Ann Arbor, MI 48104, USA. [45]Unit of Biomedicine, Bio4Dreams-Business Nursery for Life Sciences, Milano 20121, Italy. [46]Longitudinal Study Section, National Institute on Aging, Baltimore 21224, USA. [47]Institute for Maternal and Child Health - IRCCS "Burlo Garofolo", Trieste 34137, Italy. [48]Medicine, University of Colorado Anschutz Medical Campus, Aurora 80045, USA. [49]IBE, Ludwig-Maximilians-University Munich, LMU Munich, Munich 81377, Germany. [50]Institute of Medical Biostatistics, Epidemiology and Informatics (IMBEI), University Medical Center, Johannes Gutenberg University, Mainz 55101, Germany. [51]Institute of Genetic Epidemiology, Helmholtz Zentrum München - German Research Center for Environmental Health, Neuherberg 85764, Germany. [52]Program in Medical and Population Genetics, Broad Institute of MIT and Harvard, Cambridge 02142, USA. [53]Department of Epidemiology and Biostatistics, Imperial College London, London W2 1PG, UK. [54]Department of Cardiology, Ealing Hospital, London North West University Healthcare NHS Trust, Middlesex UB1 3HW, UK. [55]Vth Department of Medicine (Nephrology, Hypertensiology, Rheumatology, Endocrinology, Diabetology), Medical Faculty Mannheim, University of Heidelberg, Mannheim 68167, Germany. [56]SYNLAB MVZ Humangenetik Mannheim, Mannheim 68163, Germany. [57]Pediatrics, The Institute for Translational Genomics and Population Sciences, The Lundquist Institute for Biomedical Innovation at Harbor-UCLA, Torrance 90502, USA. [58]Department of Pediatrics, Harbor-UCLA Medical Center, Torrance 90502, USA. [59]Institute for Biomedicine, Eurac Research, Bolzano 39100, Italy. [60]Radboud university medical center, Nijmegen 6500 HB, the Netherlands. [61]Department of Internal Medicine, section Gerontology and Geriatrics, Leiden University Medical Center, Leiden 2300 RC, the Netherlands. [62]Department of Psychiatry, Amsterdam Public Health, Amsterdam UMC, Amsterdam UMC, Vrije Universiteit, Amsterdam, Amsterdam 1081 HL, the Netherlands. [63]Centre for Cardiovascular Science, Queen's Medical Research Institute, University of Edinburgh, Edinburgh EH16 4TJ Scotland, UK. [64]The Beijer laboratory and Department of Immunology, Genetics and Pathology, Uppsala University and Science for Life Laboratory, Uppsala 75237, Sweden. [65]Institute of Clinical Molecular Biology, Christian-Albrechts-University of Kiel, Kiel 24105, Germany. [66]Department of Cardiology, Leiden University Medical Center, Leiden, ZA 2333, the Netherlands. [67]Department of Epidemiology, Erasmus Medical Center, Rotterdam 3015GD, the Netherlands. [68]Institute of Genetics and Biomedic Research (IRGB), Italian National Research Council (CNR), Monserrato, (CA) 9042, Italy. [69]Center for Statistical Genetics, University of Michigan, Ann Arbor 48109, USA. [70]Singapore Eye Research Institute, Singapore National Eye Centre, Singapore 169856, Singapore. [71]Interfaculty Institute for Genetics and Functional Genomics, University Medicine Greifswald, Greifswald 17475, Germany. [72]DZHK (German Center for Cardiovascular Research), Partner Site Greifswald, Greifswald 17475, Germany. [73]Saw Swee Hock School of Public Health, National University Health System and National University of Singapore,

Singapore 117549, Singapore. [74]Department of Epidemiology, University of Groningen, University Medical Center Groningen, Groningen 9700 RB, The Netherlands. [75]Cardiovascular Epidemiology Unit, Department of Public Health and Primary Care, University of Cambridge, Cambridge CB2 OSL, UK. [76]Department of Public Health, Aarhus University, Aarhus C 8000, Denmark. [77]Division of Research, Kaiser Permenente of Northern California, Oakland 94612, USA. [78]The Scripps Research Institute, La Jolla 10550, USA. [79]National Institute on Aging, Bethesda MSC2292, USA. [80]Department of Epidemiology, Human Genetics, and Environmental Sciences, University of Texas Health Science Center at Houston, School of Public Health, Houston 77030, USA. [81]National Heart and Lung Institute, Imperial College London, London W12 0NN, UK. [82]Center for Clinical Research and Prevention, Bispebjerg and Frederiksberg Hospital, Copenhagen 2400, Denmark. [83]Department of Clinical Medicine, Faculty of Health and Medical Sciences, University of Copenhagen, Copenhagen 2200, Denmark. [84]Department of Preventive Medicine, The Icahn School of Medicine at Mount Sinai, New York 10029, USA. [85]Institute of Cardiovascular and Medical Sciences, University of Glasgow, Glasgow G12 8TA, UK. [86]Departments of Medicine, Epidemiology and Health Systems and Population Health, University of Washington, Seattle 98195, USA. [87]Medicine and Epidemiology, University of Washington, Seattle 98195, USA. [88]Faculty of Medicine, School of Health Sciences, University of Iceland, Reykjavik 101, Iceland. [89]Department of Medicine, Landspitali—The National University Hospital of Iceland, Reykjavik 101, Iceland. [90]School of Engineering and Natural Sciences, University of Iceland, Reykjavik 101, Iceland. [91]Centre for Research in Epidemiology and Population Health, Institut national de la santé et de la recherche médicale, Villejuif 94800, France. [92]UMRS 1018, University Versailles Saint-Quentin-en-Yvelines, Versailles 78035, France. [93]UMRS 1018, University Paris Sud, Villejuif 94807, France. [94]Genetic Epidemiology Unit, Dept. of Epidemiology, Erasmus University Medical Center, Rotterdam 3000CA, Netherlands. [95]Cardiology Section, Massachusetts general hospital,, Boston 02130, USA. [96]Department of Cardiology, Heart Center, Tampere University Hospital, Tampere FI-33521, Finland. [97]Department of Cardiology, Finnish Cardiovascular Research Center - Tampere, Faculty of Medicine and Health Technology, Tampere University, Tampere FI-33014, Finland. [98]Department of Genetics, University of North Carolina, Chapel Hill, NC 27599, USA. [99]Deutsches Zentrum für Herz- und Kreislauferkrankungen (DZHK), Partner Site Munich Heart Alliance, Munich 80636, Germany. [100]Region Västra Götaland, Geriatric Medicine, Institute of Medicine, Sahlgrenska Academy, University of Gothenburg, Mölndal 43180, Sweden. [101]Mary McKillop Institute for Health Research, Australian Catholic University, Melbourne 3000, Australia. [102]Department of Psychology and Logopedics, University of Helsinki, Helsinki 00014, Finland. [103]Division of Biostatistics, Washington University, St. Louis, MO 63110, USA. [104]Preventive Medicine, Northwestern University, Chicago, IL 60611, USA. [105]Department of Cardiology, Interventional Electrocardiology and Hypertension, Jagiellonian University Medical College, Kraków 31-008, Poland. [106]Geriatric Unit, Unità sanitaria locale Toscana Centro, Florence 50142, Italy. [107]Cardiovascular Department, "Ospedali Riuniti and University of Trieste", Trieste 34149, Italy. [108]Institute of Human Genetics, Klinikum rechts der Isar, Technische Universität München, München 81675, Germany. [109]Institute of Human Genetics, Helmholtz Zentrum München - German Research Center for Environmental Health, Neuherberg 85764, Germany. [110]Research Unit Molecular Epidemiology, Helmholtz Zentrum München - German Research Center for Environmental Health, Neuherberg 85764, Germany. [111]Institute of Epidemiology, Helmholtz Zentrum München - German Research Center for Environmental Health, Neuherberg 85764, Germany. [112]German Centre for Cardiovascular Research (DZHK), partner site: Munich Heart Alliance, Munich 80802, Germany. [113]Department of Cardiology, University Hospital, LMU Munich, Munich 81377, Germany. [114]Chair of Genetic Epidemiology, IBE, Faculty of Medicine, LMU Munich, Munich 81377, Germany. [115]Department of Internal Medicine, Clinical Sciences, Lund University and Skåne University Hospital, Malmo 221 85, Sweden. [116]Lund University Diabetes Center, Lund University, Malmö 221 85, Sweden. [117]Department of Clinical Epidemiology, Leiden University Medical Center, Leiden 2300 RC, the Netherlands. [118]Biological Psychology, EMGO+ Institute for Health and Care Research and Neuroscience Campus Amsterdam, VU University, Amsterdam 1081 BT, the Netherlands. [119]Department of Immunology, Genetics and Pathology, Uppsala University, Uppsala 75108, Sweden. [120]Centre for Global Health Research, Usher Institute, University of Edinburgh, Edinburgh EH8 9AG Scotland, UK. [121]Department of Medical Sciences, Cardiovascular Epidemiology, Uppsala University Hospital, Uppsala 75237, Sweden. [122]Institute of Health and Wellbeing, Faculty of Medicine, University of Glasgow, Glasgow G12 0XH, UK. [123]Laboratory of Cardiovascular Sciences, Intramural Research Program, National Institute on Aging, National Institutes of Health, Baltimore, MD 21224, USA. [124]Department of Internal Medicine B, University Medicine Greifswald, Greifswald 17475, Germany. [125]Department of Medicine, Yong Loo Lin School of Medicine, National University of Singapore, Singapore 119228, Singapore. [126]Department of Psychiatry, University of Groningen, University Medical Center Groningen, Groningen 9700 RB, The Netherlands. [127]Department of Public Health and Caring Sciences, Molecular Geriatrics, Uppsala University, Uppsala 75237, Sweden. [128]Genetic and Genomic Epidemiology Unit, Wellcome Trust Centre for Human Genetics, University of Oxford, Oxford OX3 7BN, UK. [129]Centre for Population Health Research, University of Turku and Turku University Hospital, Turku FI-20521, Finland. [130]Research Centre of Applied and Preventive Cardiovascular Medicine, University of Turku, Turku FI-20521, Finland. [131]Department of Clinical Physiology and Nuclear Medicine, Turku University Hospital, Turku FI-20521, Finland. [132]Icelandic Heart Association, Kopavogur 201, Iceland. [133]MRC Integrative Epidemiology Unit, University of Bristol, Bristol BS8 2BN, UK. [134]Population Health Sciences, Bristol Medical School,, University of Bristol, Bristol BS8 2BN, UK. [135]NIHR Barts Biomedical Research Centre, Barts and The London Faculty of Medicine and Dentistry, Queen Mary University of London, London EC1M 6BQ, UK. [136]Population Health Research Institute, St George's, University of London, London SW17 0RE, UK. [137]The Mindich Child Health and Development Institute, The Icahn School of Medicine at Mount Sinai, New York 10029, USA. [138]Department of Epidemiology, University of Washington, Seattle 98195, USA. [139]Department of Medicine, Internal Medicine, Lausanne University hospital, Lausanne 1015, Switzerland. [140]MRC Human Genetics Unit, Institute of Genetics and Cancer, University of Edinburgh, Edinburgh EH4 2XU Scotland, UK. [141]Department of Metabolism, Imperial College London, London W12 0HS, UK. [142]Inserm/CNRS UMR 1283/8199, Pasteur Institute of Lille, Lille University Hospital, EGID, Lille 59000, France. [143]University of Lille, Lille 59000, France. [144]Institute for Molecular Medicine Finland (FIMM), University of Helsinki, Helsinki 00290, Finland. [145]Cardiology Section, VA Boston Healthcare System, Harvard Medical School, Boston, MA 02132, USA. [146]Department of Clinical Physiology, Tampere University Hospital, Tampere FI-33521, Finland. [147]Department of Clinical Physiology, Finnish Cardiovascular Research Center - Tampere, Faculty of Medicine and Health Technology, Tampere University, Tampere FI-33521, Finland. [148]Institute for Cardiogenetics, University of Lübeck, Lübeck 23562, Germany. [149]Department of Drug Treatment, Sahlgrenska University Hospital, Gothenburg 41345, Sweden. [150]Centre for Genomic and Experimental Medicine, Institute of Genetics & Molecular Medicine, University of Edinburgh, Edinburgh EH4 2XU, UK. [151]Department of General practice and primary care, University of Helsinki, Helsinki 00014, Finland. [152]Department of Obstetrics and Gynecology, National University of Singapore, Singapore 119228, Singapore. [153]Public health Research Program, Folkhalsan Research Center, Helsinki 000250, Finland. [154]Centre for Urban Epidemiology, University Hospital of Essen, University Duisburg-Essen, Essen 45122, Germany. [155]Epidemiology, Harvard T.H. Chan School of Public Health, Boston, MA 02112, USA. [156]Institute of Biomedical Technologies, National Research Council of Italy, Segrate, (MI) 20090, Italy. [157]Department of Medicine, Surgery and Health Sciences, University of Trieste, Trieste 34149, Italy. [158]Jackson Heart Study, University of Mississippi Medical Center, Jackson 39216, USA. [159]Chair of Epidemiology, Institute for Medical Information Processing, Biometry and Epidemiology, Ludwig-Maximilians-Universität München, Munich 81377, Germany. [160]Lee Kong Chian School of Medicine, Nanyang Technological University, Singapore 308232, Singapore. [161]Imperial College Healthcare NHS Trust, Imperial College London, London W12 0HS, UK. [162]Synlab Academy, Synlab Holding Deutschland GmbH, Mannheim 68161, Germany. [163]Department of Cardiology, Clinical Sciences, Lund University and Skåne University Hospital, Lund 221 85, Sweden. [164]Wallenberg Center for Molecular Medicine and Lund University Diabetes Center, Lund University, Lund 221 84, Sweden. [165]The Wallenberg Laboratory/Department of Molecular and Clinical Medicine, Institute of Medicine, Gothenburg University

and the Department of Cardiology, Sahlgrenska University Hospital, Gothenburg 413 45, Sweden. [166]Department of Public Health and Primary Care, Leiden University Medical Center, Leiden 2300 RC, the Netherlands. [167]MRC Biostatistics Unit, University of Cambridge, Cambridge CB2 0SR, UK. [168]Institute of Epidemiology and Biobank PopGen, Kiel University, Kiel 24105, Germany. [169]Einthoven Laboratory for Experimental Vascular Medicine, Leiden University Medical Center, Leiden, ZA 2333, the Netherlands. [170]Netherlands Heart Institute, Utrecht 3511 EP, the Netherlands. [171]Department of Nephrology, University Medical Center Groningen, Groningen 9700RB, the Netherlands. [172]Ophthalmology & Visual Sciences Academic Clinical Program (Eye ACP), Duke-NUS Medical School, Singapore 169857, Singapore. [173]Department of Ophthalmology, Yong Loo Lin School of Medicine, National University of Singapore, Singapore 119228, Singapore. [174]Tsinghua Medicine, Tsinghua University, Beijing 100084, China. [175]Interdisciplinary Center Psychopathology and Emotion Regulation, University of Groningen, University Medical Center Groningen, Groningen 9700 RB, The Netherlands. [176]Department of Genetics, University of Groningen, University Medical Center Groningen, Groningen 9700RB, the Netherlands. [179]These authors contributed equally: Yordi J. van de Vegte, Ruben N. Eppinga. ✉e-mail: P.vanderHarst@umcutrecht.nl

## The DCCT/EDIC Research Group

**Delnaz Roshandel[177] & Andrew D. Paterson[177,178]**

[177]Genetics and Genome Biology Program, The Hospital for Sick Children, Toronto M5G 0A4, Canada. [178]Dalla Lana School of Public Health, University of Toronto, Toronto, ON M5T 3M7, Canada.

A list of members and their affiliations appears in the Supplementary Information.

