## [Peer Review File · Nature Communications]

Genetic insights into resting heart rate and its role in cardiovascular diseaseReviewers' comments:

Reviewer #1 (Remarks to the Author):

P. van den Harst and colleagues report a large multi-national meta-analysis of genome-wide association studies on resting heart rate, including in total more than 835,000 individuals.

They identified in total 493 independent signals at 352 loci, across which they prioritized 670 candidate genes based on multiple in silico analyses. They showed that these genes were mostly related to cardiac biology, and that the associated variants exert the largest effect on ventricular repolarization based on ECG-wide Mendelian randomization.

Finally, by two-sample Mendelian randomization, they report that increased genetically predicted RHR is associated with increased risk of dilated cardiomyopathy and decreased risk of atrial fibrillation and stroke. Of particular note, they also report no evidence for genetic association between RHR and all-cause mortality, which contradicts their previous report of an initial GWAS based on 265,046 individuals (PMID: 27798624)

This study is undoubtedly the largest genome-wide association meta-analysis on RHR reported to date. It provides additional clues towards understanding RHR biology. I have however some concerns on the study design, on the novelty of the GWAS results, and on the robustness of downstream MR analyses.

In the abstract, the authors state that they have performed a meta-analysis of 100 GWAS's on RHR. In reality, they have performed two independent GWAS's: one large meta-analysis including GWAS's based on 99 distinct cohorts for a total of 351k subjects (IC-RHR); another large GWAS based on 484k individuals from the UKBB.

The authors should note that the GWAS on RHR based on UKBB has already been published elsewhere (PMID: 31648709). The authors of this previous study reported 437 independent loci, based on the analysis of 428k selected individuals.

Digging into the characteristics of the IC-RHR cohorts (reported in Supp. Data 1), these cohorts are highly heterogeneous, in terms of size and RHR measurements. For example, RHR are reported as quantitative values for all cohorts except one: 23andMe reports RHR as a self-reported categorical trait (" <50 bpm: $n=51039$; $50-59$ bpm: $n=9843$; $60-69$ bpm: $n=19255$; $70-79$ bpm: $n=13161$; $80-89$ bpm: $n=5277$; >90 bpm: $n=1510$ " - note that there is an error in the number of subjects reporting a RHR below 50 bpm, which might be 1,993 and not 51,039). Moreover, it seems that 21 of the 99 cohorts include very low numbers of subjects (less than 1,000 in total) while 2 cohorts include more than 50,000 subjects (23andMe; deCODE, which include first/second degree relatives). Such high heterogeneity may introduce substantial biases in the results of the IC-RHR meta-analysis (Supp. Data 2). This should be highlighted as a limitation of this study.

Rather than presenting this study as one large meta-analysis, the authors could present their result as a thorough replication of the UKBB GWAS results, based on the exploitation of the heterogeneous set of

IC-RHR cohorts. This design would reassure the reader on the robustness of the results obtained on UKBB. Following this approach, the authors would only report and further consider the replicated signals, which would be more convincing for the reader.

In addition of the Supp. Data 2, the authors should include a summary table with the list of replicated loci, the candidate genes prioritized for each locus, the number of associated signals, the top signal in UKBB and its replication in IC-RHR. This would be of great help for the reader.

« A total of 407 unique genes were in close proximity to the lead variant, 327 defined as the nearest gene and any additional gene within 10kb »: Fixing a threshold of 10 kb for gene proximity seems very conservative. The authors should consider increasing this threshold (e.g. up to 100kb, which would still be relatively conservative).

The authors performed two-sample MR analysis to test whether genetically predicted RGR is associated with all-cause mortality, and found no significant signal. This result is in direct contradiction with their previous report entitled « Identification of genomic loci associated with resting heart rate and shared genetic predictors with all-cause mortality » (PMID: 27798624). Then the authors investigated the most likely cause for such discrepancy and concluded that previous one-sample MR analyses led to false positive findings.

This important result points that such MR analyses should be considered with caution, and should be consistently replicated by independent studies before any definitive conclusion. The authors should further highlight this point in the abstract of the manuscript, and report more explicitly the major discrepancies with their previous findings.

Minor comments:

There is no reference to the previous work published by the authors based on the Exome Chip (PMID: 28379579). Did they confirm the RHR loci identified by this earlier work?

There are remaining typos or grammatical errors across the manuscript: it should be carefully reviewed before any re-submission.

Examples:

Abstract: 'We prioritize 670 genes..' should be 'We prioritized...', and the corresponding sentence should be revised

Figure 1B: 'triats' should be 'traits'

Some references to Figures are not formatted correctly across the manuscript (no number). This should be checked.

Reviewer #2 (Remarks to the Author):

An extensive investigation to explore possible causative relationship between resting heart rate, disease, and mortality.

Strengths include the large data set incorporating data from 484,307 UK Biobank participants and 351,158 samples from 99 cohorts as part of IC-RHR data, use of 2-sample MR and DEPICT pathway and tissue analysis.

In contrast to previous reports, there was no evidence for either a linear or U-shaped association between genetically determined RHR and total mortality or disease specific mortality. This is the important finding and supports the concept that RHR reflects physical fitness and a number of comorbidities affecting life expectancy.

Other reported outcomes include a relationship between higher RHR and dilated cardiomyopathy. Here clarification is needed for the UK Biobank definition of 'dilated cardiomyopathy', a clinical diagnosis with many etiologies. Similarly, how was 'heart failure' defined and does this include HF_rEF and HF_pEF?

Supp Fig 7 suggests that the relationship between genetically determined RHR and dilated cardiomyopathy' is driven by a small number of individual SNPs.

The presentation of the findings related to genetic RHR and atrial fibrillation are confusing. Indeed, a significant negative relationship is reported in the abstract but such a relationship was only significant using the MR Lasso approach and is not defined as significant in the discussion. Since the authors find a significant negative relationship between genetically determined RHR and cardio-embolic stroke, an AF relationship is plausible given that AF is the major etiological factor in this stroke subtype. It should be noted that atrial fibrillation is often a missed diagnosis.

To consider adding to the discussion on possible mechanisms, endurance athletes are known to have substantially greater risk of AF in association with very low RHR.

Minor comments

The results and discussion are somewhat iterative. Suggest editing and shortening to improve readability

Point-by-point response to reviewers

We thank the reviewers for their comments on our manuscript. We shall address the comments sequentially below.

Reviewer #1:

P. van den Harst and colleagues report a large multi-national meta-analysis of genome-wide association studies on resting heart rate, including in total more than 835,000 individuals. They identified in total 493 independent signals at 352 loci, across which they prioritized 670 candidate genes based on multiple in silico analyses. They showed that these genes were mostly related to cardiac biology, and that the associated variants exert the largest effect on ventricular repolarization based on ECG-wide Mendelian randomization. Finally, by two-sample Mendelian randomization, they report that increased genetically predicted RHR is associated with increased risk of dilated cardiomyopathy and decreased risk of atrial fibrillation and stroke. Of particular note, they also report no evidence for genetic association between RHR and all-cause mortality, which contradicts their previous report of an initial GWAS based on 265,046 individuals (PMID: 27798624). This study is undoubtedly the largest genome-wide association meta-analysis on RHR reported to date. It provides additional clues towards understanding RHR biology. I have however some concerns on the study design, on the novelty of the GWAS results, and on the robustness of downstream MR analyses.

Response: We thank the reviewer for the response and will discuss the concerns raised below.

In the abstract, the authors state that they have performed a meta-analysis of 100 GWAS's on RHR. In reality, they have performed two independent GWAS's: one large meta-analysis including GWAS's based on 99 distinct cohorts for a total of 351k subjects (IC-RHR); another large GWAS based on 484k individuals from the UKBB. The authors should note that the GWAS on RHR based on UKBB has already been published elsewhere (PMID: 31648709). The authors of this previous study reported 437 independent loci, based on the analysis of 428k selected individuals.

Response: We are aware that a GWAS on resting heart rate has already been performed within the UK Biobank, but we disagree that this decreases the novelty of the GWAS results as stated in the introduction. We almost double the sample size of the current study ($n = 835,465$) compared to the study from Guo *et al.* ($n = 428,250$)¹. GWAS of blood pressure^{2,3} and atrial fibrillation^{4,5} had already been published in large populations as well before publication of even larger meta-analyses, which increased their statistical power. This increase in sample size allowed us to alter our GWAS methodology on several key points from the study from Guo *et al.* We erred on the conservative side compared to the study from Guo *et al.* by adopting a more stringent LD threshold ($R^2 < 0.005$ versus $R^2 < 0.1$, respectively) and P -value threshold ($P < 1 \times 10^{-8}$ versus $P < 5 \times 10^{-8}$), in order to ensure robust findings and minimize type-I error rates even for lower frequency variants⁶. In addition, the increase in sample size enabled us to perform an one-stage replication analysis to bring forward internally replicated loci robustly associated with RHR, which has not been performed on this scale in any previous RHR GWAS^{1,7,8}. We added **Supplementary Table 3** to make an in-depth comparison with the results from previous GWAS on resting heart rate performed by Guo *et al.*, Eppinga *et al.* and Den Hoed *et al.* We added the following sentences to the manuscript:

“Out of these 493 independent genetic variants, 68 were outside previously identified RHR associated loci and 67 of those were internally replicated (Figure 2A, Supplementary Data 2)^{1,8,9}. Out of the 425 genetic variants inside previously identified RHR associated loci, a total of 376 were internally replicated.”

And:

“The RHR associated genetic variants identified previous studies from Eppinga et al. and Den hoed et al. were all replicated in the current study (Supplementary Data 3). A total of 74 loci identified in the study from Guo et al. were not replicated in the current study, of which 40 would not have been identified as locus using the current GWAS clumping criteria. The remaining 34 loci did not reach genome-wide significance in the current meta-analysis with generally high P-values in the IC-RHR consortium, probably therefore failing replication (Supplementary Data 3).”

And the following sentence to the discussion to highlight the importance of increasing the sample size for the current GWAS meta-analysis:

“This increase of samplesize allowed us to report 68 novel RHR associated genetic variants and, importantly, provide internal replication for 376 genetic variants previously associated with RHR.”

Digging into the characteristics of the IC-RHR cohorts (reported in Supp. Data 1), these cohorts are highly heterogeneous, in terms of size and RHR measurements. For example, RHR are reported as quantitative values for all cohorts except one: 23andMe reports RHR as a self-reported categorical trait (“<50 bpm: n=51039; 50-59 bpm: n=9843; 60-69 bpm: n=19255; 70-79 bpm: n=13161; 80-89 bpm: n=5277; >90 bpm: n=1510” - note that there is an error in the number of subjects reporting a RHR below 50 bpm, which might be 1,993 and not 51,039). Moreover, it seems that 21 of the 99 cohorts include very low numbers of subjects (less than 1,000 in total) while 2 cohorts include more than 50,000 subjects (23andMe; deCODE, which include first/second degree relatives). Such high heterogeneity may introduce substantial biases in the results of the IC-RHR meta-analysis (Supp. Data 2). This should be highlighted as a limitation of this study.

Response: The concern raised states that the IC-RHR cohorts are highly heterogeneous in sample size and measurement of RHR. A downstream result would be a high heterogeneity in the SNP-statistics of our GWAS, but we find that our Chi-square test-statistics¹⁰ indicate differently. A total of 493 independent RHR associated genetic variants were brought forward in the current study, of which 28 were heterogeneous within the IC-RHR cohort at a P-value cut-off of 0.05, and only 8 at a P-value cut-off of 0.01. These results are added to **Supplementary Data 2**, columns “BB-BD”.

Heterogeneous sample sizes only introduce bias if unweighted meta-analysis are performed. However, the current meta-analysis uses standard errors to correctly account for the sample size of the cohort¹⁰. Sample sizes of previous GWAS meta-analyses vary widely as well³⁻⁵, which is actually a reason to perform a meta-analysis on this type of data.

The RHR measurements have been performed using different, but valid methodology. The self-reported categorical RHR measurement of 23andMe has already been used in our previous study published in Nature Genetics⁸, and should uphold the standards set by Nature papers.

Rather than presenting this study as one large meta-analysis, the authors could present their result as a thorough replication of the UKBB GWAS results, based on the exploitation of the heterogeneous set of IC-RHR cohorts. This design would reassure the reader on the robustness of the results obtained on UKBB. Following this approach, the authors would only report and further consider the replicated signals, which would be more convincing for the reader.

Response: The comment of the reviewer builds further on the first two concerns, which have been independently discussed above. In summary, we do not aim to solely replicate the findings from Guo *et al.*, because we a) adopt a more appropriate and stringent LD and *P*-value threshold to decrease type I error rates, b) already perform an internal replication, and c) provide evidence that the variants discovered in our IC-RHR cohort are not highly heterogenic (**Supplementary Data 2**).

In addition of the Supp. Data 2, the authors should include a summary table with the list of replicated loci, the candidate genes prioritized for each locus, the number of associated signals, the top signal in UKBB and its replication in IC-RHR. This would be of great help for the reader.

Response: These were already present in **Supplementary Data 2** during the first submission. The updated location of this information in **Supplementary Data 2** is columns “I-K” for replicated loci, columns “AC-AF” for candidate genes, the row index for the number of associated signals, and columns “U”, “AN”, and “AX” for the *P*-values of the full meta-analysis, the UK Biobank, and the meta-analysis of the IC-RHR consortium, respectively. Please note that this is a rather large table of which the left side is locked to keep the identification of the genetic variants on the left side when scrolling sideways, this can of course be disabled if preferred for browsing.

A total of 407 unique genes were in close proximity to the lead variant, 327 defined as the nearest gene and any additional gene within 10kb »: Fixing a threshold of 10 kb for gene proximity seems very conservative. The authors should consider increasing this threshold (e.g. up to 100kb, which would still be relatively conservative).

Response: The 10kb window was chosen based on our previous GWAS on RHR⁸. This cut-off is in line with recent large scale meta-analyses published in Nature journals and should uphold to current GWAS standards^{11,12}.

The authors performed two-sample MR analysis to test whether genetically predicted RGR is associated with all-cause mortality, and found no significant signal. This result is in direct contradiction with their previous report entitled « Identification of genomic loci associated with resting heart rate and shared genetic predictors with all-cause mortality » (PMID: 27798624). Then the authors investigated the most likely cause for such discrepancy and concluded that previous one-sample MR analyses led to false positive findings.

This important result points that such MR analyses should be considered with caution, and should be consistently replicated by independent studies before any definitive conclusion. The authors should further highlight this point in the abstract of the manuscript, and report more explicitly the major discrepancies with their previous findings.

Response: The reviewer expressed concerns on the robustness of the MR analyses in the introduction of the review. However, the above mentioned comment does not provide any elaboration on possible methodological or statistical issues that might have raised this concern. The reviewer only points to the discrepant results of our current study compared to our previous study⁸ and denotes that careful interpretation and replication is warranted. We actually agree with the reviewer that all scientific results should be interpreted with caution and replicated if possible, and this is one of the reasons for which we sought out to replicate the results of our previous study. Unfortunately, we were unable to replicate these findings, but the chance of finding discrepant results is unavoidable and the core reason for the existence of replication studies in the first place.

We performed extensive sensitivity analysis to pinpoint the cause of this discrepancy and find that the positive relationship discovered in our previous study is most likely due to an increased type I error rate introduced by weak-instrument bias in an one-sample MR setting, a possible methodological issue well-

described in literature¹³. The reviewer proceeds to state that we should highlight the major discrepancies with our previous findings, but our systematic alteration of key differences with previous results already highlights this (please see **Methods** on page 28 and 29 and in the figure caption of **Supplementary Figure 5**). We agree that the cause of the discrepant results should be added to the abstract and changed the following sentence:

“We found no evidence for a linear or non-linear genetic association between RHR and all-cause mortality.”

To:

“We found no evidence for a linear or non-linear genetic association between RHR and all-cause mortality in contrast to our previous Mendelian randomization study. Systematic alteration of key differences between the current and previous Mendelian randomization study indicated that the most likely cause of the discrepancy between these studies arises from false positive findings in previous one-sample MR analyses caused by weak-instrument bias at lower P-value thresholds.”

We strongly believe that the current paper provides as robust MR results as possible to ensure nuanced claims on the genetic association between RHR and all-cause mortality, as we use current gold standard methodology including a two-sample MR approach and state-of-the art MR techniques¹⁴.

Minor comments:

There is no reference to the previous work published by the authors based on the Exome Chip (PMID: 28379579). Did they confirm the RHR loci identified by this earlier work?

Response: Based on the comments from the editor we have added a full comparison with the study from Den Hoed *et al.* The study referred to replicates the RHR associated loci from the study from Den Hoed *et al.*, the current study replicates these loci as well. Further comparison with the study from van den Berg *et al.* was not performed, considering a genome-wide and not an exome-wide association study was performed. However, we added a full comparison with a more recent study from Guo *et al.* the largest genome-wide association study on resting heart rate to date.

There are remaining typos or grammatical errors across the manuscript: it should be carefully reviewed before any re-submission.

Examples:

Abstract: ‘We prioritize 670 genes..’ should be ‘We prioritized...’, and the corresponding sentence should be revised

Figure 1B: ‘triats’ should be ‘traits’

Response: We corrected several remaining typos and grammatical errors.

Some references to Figures are not formatted correctly across the manuscript (no number). This should be checked.

Response: We checked all references to (Supplementary) Figures, Tables and Data and changed some minor errors. Please also note that some Figures and Tables are referred to in text as a range, such as “**Supplementary Figures 7-11**”.

Reviewer #2:

An extensive investigation to explore possible causative relationship between resting heart rate, disease, and mortality.

Strengths include the large data set incorporating data from 484,307 UK Biobank participants and 351,158 samples from 99 cohorts as part of IC-RHR data, use of 2-sample MR and DEPICT pathway and tissue analysis.

In contrast to previous reports, there was no evidence for either a linear or U-shaped association between genetically determined RHR and total mortality or disease specific mortality. This is the important finding and supports the concept that RHR reflects physical fitness and a number of comorbidities affecting life expectancy.

Response: We thank the reviewer for the overall positive response to our study.

Other reported outcomes include a relationship between higher RHR and dilated cardiomyopathy. Here clarification is needed for the UK Biobank definition of ‘dilated cardiomyopathy’, a clinical diagnosis with many etiologies. Similarly, how was ‘heart failure’ defined and does this include HF_rEF and HF_pEF?

Response: We thank the reviewer for the question. Dilated cardiomyopathy (DCM) was defined according to ICD-10 code I420 and ICD-9 code 4254. This includes most likely familial dilated cardiomyopathies, as it excludes other causes such as ischemia, alcohol, drugs, as well as post-partum and eosinophilic cardiomyopathies. The same applies to hypertrophic cardiomyopathy (HCM), which was only based on ICD-10 codes I420 and I422, as well as ICD-9 code 4251. Heart failure was defined as a compound definition of all types of heart failure, whereas heart failure excluding cardiomyopathies was defined as the compound definition for heart failure excluding those for DCM and HCM (please see **Supplementary Data 15** for the full overview). We were able to differentiate between HF_rEF and HF_pEF in our familial DCM and HCM definition, but unfortunately not in the compound definitions of heart failure due to the unavailability of data on left ventricular ejection fraction and function.

We added the following sentence to the discussion:

“Our MR analysis on the compound definition of heart failure could be hampered by its phenotypical heterogeneity, as we were unable to differentiate between heart failure with reduced and preserved ejection fraction. It would be interesting to repeat current Mendelian randomization analysis if more in-depth phenotyping on left ventricular ejection fraction and function becomes available, especially considering the different effects of RHR on familial dilated versus hypertrophic cardiomyopathy in the current study.”

Supp Fig 7 suggests that the relationship between genetically determined RHR and dilated cardiomyopathy’ is driven by a small number of individual SNPs.

Response: We thank the reviewer for the response. The scatterplot does show several genetic variants strongly associated with RHR that might drive the association between RHR and dilated cardiomyopathy. However, outlier robust sensitivity analyses show results similar to the main inverse-variance weighted multiplicative random effects model. This consistency of the MR results make it more likely that the results are not driven by a small number of SNPs, as these SNP effects would otherwise have been downweighed in the sensitivity analyses resulting in a null-effect.

The presentation of the findings related to genetic RHR and atrial fibrillation are confusing. Indeed, a significant negative relationship is reported in the abstract but such a relationship was only significant using the MR Lasso approach and is not defined as significant in the discussion. Since the authors find a significant negative relationship between genetically determined RHR and cardio-embolic stroke, an AF relationship is plausible given that AF is the major etiological factor in this stroke subtype. It should be noted that atrial fibrillation is often a missed diagnosis. To consider adding to the discussion on possible mechanisms, endurance athletes are known to have substantially greater risk of AF in association with very low RHR.

Response: We thank the reviewer for the response. Our conclusion on the significant negative association between RHR and atrial fibrillation is predominantly based on the non-linear MR which showed a negative exponential dose response curve. However, we agree that this is difficult to distillate from the current text and we therefore changed the discussion section on atrial fibrillation on page 17 and added the following sentences to the discussion:

“The linear MR between RHR and atrial fibrillation provided suggestive evidence for an inverse relationship between RHR and atrial fibrillation, in line with a previous linear MR study¹⁵. We do find a significant negative exponential dose-response curve between RHR and atrial fibrillation in support of an inverse relationship, and take the non-linear MR forward as the main result considering the fractional polynomial test indicated that a non-linear model fitted the localized average causal effect estimates better than the linear model. Previous observational studies on the relationship between RHR and atrial fibrillation have shown conflicting results and have described various relationships including inverse linear¹⁶⁻¹⁸, U-shaped¹⁸ and J-shaped¹⁹ associations. All these association patterns support the hypothesis that individuals with a low RHR might exhibit a higher risk of atrial fibrillation development compared to those with an average RHR. A recent stratified Mendelian randomization showed an inverse genetic relationship between RHR and atrial fibrillation in individuals with a RHR below 65 bpm as well²⁰. Possible mechanisms that could underly an increased risk of atrial fibrillation in individuals with a low RHR include increased left atrial stroke volume and consequent atrial remodeling due to myocyte stretching²¹, or an increased vagal tone promoting global disorganization in the left atrium due to increased heterogeneity of the refractory period²². In contrast to the often hypothesized U-shaped or J-shaped association^{18,19}, we find a decreasing risk of atrial fibrillation development in those with a high RHR. One potential explanation is that previous observational studies were affected by collider bias through confounding factors which increase atrial fibrillation risk and typically occur in tandem with a high rather than a low RHR, such as hypertension²³ and obesity²⁴. We advocate for cautious interpretation of current result due to the diverse biological mechanisms through which the RHR associated genetic variants alter the risk of atrial fibrillation development⁷.”

Minor comments:

The results and discussion are somewhat iterative. Suggest editing and shortening to improve readability

Response: We agree that the manuscript is rather iterative. We tried to shorten parts of the manuscript.

References

1. Guo, Y., Chung, W., Zhu, Z., *et al.* Genome-Wide Assessment for Resting Heart Rate and Shared Genetics With Cardiometabolic Traits and Type 2 Diabetes. *J. Am. Coll. Cardiol.* 74, 2162–2174 (2019).
2. Wain, L. V., Vaez, A., Jansen, R., *et al.* Novel Blood Pressure Locus and Gene Discovery Using Genome-Wide Association Study and Expression Data Sets From Blood and the Kidney. *Hypertens. (Dallas, Tex. 1979)* 70, e4–e19 (2017).
3. Evangelou, E., Warren, H. R., Mosen-Ansorena, D., *et al.* Genetic analysis of over 1 million people identifies 535 new loci associated with blood pressure traits. *Nat. Genet.* 50, 1412–1425 (2018).
4. Christophersen, I. E., Rienstra, M., Roselli, C., *et al.* Large-scale analyses of common and rare variants identify 12 new loci associated with atrial fibrillation. *Nat. Genet.* 49, 946–952 (2017).
5. Roselli, C., Chaffin, M. D., Weng, L. C., *et al.* Multi-ethnic genome-wide association study for atrial fibrillation. *Nat. Genet.* 50, 1225–1233 (2018).
6. Fadista, J., Manning, A. K., Florez, J. C. & Groop, L. The (in)famous GWAS P-value threshold revisited and updated for low-frequency variants. *Eur. J. Hum. Genet.* 2016 248 24, 1202–1205 (2016).
7. Den Hoed, M., Eijgelsheim, M., Esko, T., *et al.* Identification of heart rate-associated loci and their effects on cardiac conduction and rhythm disorders. *Nat. Genet.* 45, 621–631 (2013).
8. Eppinga, R. N., Hagemeijer, Y., Burgess, S., *et al.* Identification of genomic loci associated with resting heart rate and shared genetic predictors with all-cause mortality. *Nat. Genet.* 48, 1557–1563 (2016).
9. Den Hoed, M., Eijgelsheim, M., Esko, T., *et al.* Identification of heart rate-associated loci and their effects on cardiac conduction and rhythm disorders. *Nat. Genet.* 45, 621–631 (2013).
10. Willer, C. J., Li, Y. & Abecasis, G. R. METAL: fast and efficient meta-analysis of genomewide association scans. *Bioinformatics* 26, 2190–2191 (2010).
11. Evangelou, E., Warren, H. R., Mosen-Ansorena, D., *et al.* Genetic analysis of over 1 million people identifies 535 new loci associated with blood pressure traits. *Nat. Genet.* 50, 1412–1425 (2018).
12. Watanabe, K., Jansen, P. R., Savage, J. E., *et al.* Genome-wide meta-analysis of insomnia prioritizes genes associated with metabolic and psychiatric pathways. doi:10.1038/s41588-022-01124-w.
13. Burgess, S. & Thompson, S. G. Avoiding bias from weak instruments in Mendelian randomization studies. *Int. J. Epidemiol.* 40, 755–764 (2011).
14. Burgess, S., Davey Smith, G., Davies, N. M., *et al.* Guidelines for performing Mendelian randomization investigations. *Wellcome Open Res.* 4, 186 (2019).
15. Larsson, S. C., Drca, N., Mason, A. M. & Burgess, S. Resting Heart Rate and Cardiovascular Disease. *Circ. Genomic Precis. Med.* 12, e002459 (2019).
16. Thelle, D. S., Selmer, R., Gjesdal, K., *et al.* Resting heart rate and physical activity as risk factors for lone atrial fibrillation: a prospective study of 309 540 men and women. *Heart* 99, 1755–1760 (2013).
17. Morseth, B., Graff-Iversen, S., Jacobsen, B. K., *et al.* Physical activity, resting heart rate, and atrial fibrillation: the Tromsø Study. *Eur. Heart J.* 37, 2307–2313 (2016).
18. Elliott, A. D., Mahajan, R., Lau, D. H. & Sanders, P. Atrial Fibrillation in Endurance Athletes: From Mechanism to Management. *Cardiol. Clin.* 34, 567–578 (2016).
19. Liu, X., Guo, N., Zhu, W., *et al.* Resting Heart Rate and the Risk of Atrial Fibrillation A Dose-Response Analysis of Cohort Studies. *Int. Heart J.* 60, 805–811 (2019).
20. Siland, J. E., Geelhoed, B., Roselli, C., *et al.* Resting heart rate and incident atrial fibrillation: A stratified Mendelian randomization in the AFGen consortium. *PLoS One* 17, e0268768 (2022).
21. Iwasaki, Y., Nishida, K., Kato, T. & Nattel, S. Atrial Fibrillation Pathophysiology. *Circulation*

- 124, 2264–2274 (2011).
22. Kneller, J., Zou, R., Vigmond, E. J., *et al.* Cholinergic atrial fibrillation in a computer model of a two-dimensional sheet of canine atrial cells with realistic ionic properties. *Circ. Res.* 90, e73–e87 (2002).
 23. Lip, G. Y. H., Coca, A., Kahan, T., *et al.* Hypertension and cardiac arrhythmias: a consensus document from the European Heart Rhythm Association (EHRA) and ESC Council on Hypertension, endorsed by the Heart Rhythm Society (HRS), Asia-Pacific Heart Rhythm Society (APHRS) and Sociedad Latinoamericana de Estimulación Cardíaca y Electrofisiología (SOLEACE). *Europace* 19, 891–911 (2017).
 24. Overvad, T. F., Rasmussen, L. H., Skjøth, F., *et al.* Body mass index and adverse events in patients with incident atrial fibrillation. *Am. J. Med.* 126, (2013).

REVIEWERS' COMMENTS

Reviewer #1 (Remarks to the Author):

On this revised manuscript by Pim van der Harst and Colleagues, the authors have properly addressed the referees' comments. They have either edited the manuscript following the reviewers' suggestions, or have provided solid argumentation to express disagreements. Overall, the manuscript is substantially improved, and in particular better tuned to describe the novel results in light of previous reports: I have no other major comment, and congratulate the authors for this great work.

Reviewer #2 (Remarks to the Author):

The authors have made an adequate response to my previous queries. As noted, in depth phenotyping for each of the cardiomyopathy classifications would provide further biological insight.

Minor additional comment

Clearly atrial fibrillation is the major cause of cardioembolic stroke. As such, it would be important to add a sentence in the Discussion to the effect that the 'although the relationship between a low RHR and cardioembolic stroke was attenuated by only 18.4% when corrected for atrial fibrillation, this may be due to the fact that atrial fibrillation is commonly a missed diagnosis'.

Point-by-point response to reviewers

We thank the reviewers for their comments on our manuscript. We shall address the comments sequentially below.

Reviewer #1:

On this revised manuscript by Pim van der Harst and Colleagues, the authors have properly addressed the referees' comments. They have either edited the manuscript following the reviewers' suggestions, or have provided solid argumentation to express disagreements. Overall, the manuscript is substantially improved, and in particular better tuned to describe the novel results in light of previous reports: I have no other major comment, and congratulate the authors for this great work.

Response: We thank the reviewer for the overall positive assessment of our work.

Reviewer #2:

The authors have made an adequate response to my previous queries. As noted, in depth phenotyping for each of the cardiomyopathy classifications would provide further biological insight.

Response: We thank the reviewer for the overall positive assessment of the rebuttal and the manuscript. Unfortunately, the lack of echocardiographic data in the UK Biobank still prevents us in our differentiation between HFrEF and HFpEF in non-syndromic heart failure. We agree that in-depth phenotyping of heart failure might provide further biological insights and want to emphasize its importance for future studies. We hope the following sentence in the discussion accomplishes this goal:

“Our MR analysis on the compound definition of heart failure could be hampered by its phenotypical heterogeneity, as we were unable to differentiate between heart failure with reduced and preserved ejection fraction. It would be interesting to repeat current Mendelian randomization analysis if more in-depth phenotyping on left ventricular ejection fraction and function becomes available, especially considering the different effects of RHR on familial dilated versus hypertrophic cardiomyopathy in the current study.”

Minor additional comment

Clearly atrial fibrillation is the major cause of cardioembolic stroke. As such, it would be important to add a sentence in the Discussion to the effect that the ‘although the relationship between a low RHR and cardioembolic stroke was attenuated by only 18.4% when corrected for atrial fibrillation, this may be due to the fact that atrial fibrillation is commonly a missed diagnosis’.

Response: We thank the reviewer for this additional comment and agree that this should be mentioned in the manuscript. We have added the following sentence to the discussion section.

“The relationship between a low RHR and cardioembolic stroke was attenuated by only 18.4% when corrected for atrial fibrillation, which might underestimated due atrial fibrillation being a commonly a missed diagnosis.”